# Early immune markers of clinical, virological, and immunological outcomes in patients with COVID-19: a multi-omics study

Zicheng Hu[1,2]*, Kattria van der Ploeg[3], Saborni Chakraborty[3],
Prabhu S Arunachalam[4], Diego AM Mori[3], Karen B Jacobson[3], Hector Bonilla[3],
Julie Parsonnet[3,5], Jason R Andrews[3], Marisa Holubar[3], Aruna Subramanian[3],
Chaitan Khosla[6], Yvonne Maldonado[7], Haley Hedlin[8], Lauren de la Parte[3],
Kathleen Press[3], Maureen Ty[3], Gene S Tan[9,10], Catherine Blish[3,11], Saki Takahashi[12],
Isabel Rodriguez-Barraquer[12], Bryan Greenhouse[11,12], Atul J Butte[1],
Upinder Singh[3,13], Bali Pulendran[4,13,14], Taia T Wang[3,11,13], Prasanna Jagannathan[3,11]*

[1]Bakar Computational Health Sciences Institute, University of California, San Francisco, United States; [2]Department of Microbiology and Immunology, University of California, San Francisco, United States; [3]Department of Medicine, Stanford University, Stanford, United States; [4]Institute for Immunity, Transplantation, and Infection, Stanford University, Stanford, United States; [5]Department of Epidemiology and Population Health, Stanford University, Stanford, United States; [6]ChEM-H, Stanford University, Stanford, United States; [7]Department of Pediatrics, Stanford University, Stanford, United States; [8]Quantitative Sciences Unit, Stanford University, Stanford, United States; [9]J. Craig Venter Institute, San Diego, United States; [10]Division of Infectious Diseases, Department of Medicine, University of California, San Diego, United States; [11]Chan Zuckerberg Biohub, San Francisco, United States; [12]Department of Medicine, University of California, San Francisco, United States; [13]Department of Microbiology and Immunology, Stanford University, Stanford, United States; [14]Department of Pathology, Stanford University, Stanford, United States

*For correspondence:
hzc363@gmail.com (ZH);
prasj@stanford.edu (PJ)

## Abstract

**Background:** The great majority of severe acute respiratory syndrome-related coronavirus 2 (SARS-CoV-2) infections are mild and uncomplicated, but some individuals with initially mild COVID-19 progressively develop more severe symptoms. Furthermore, there is substantial heterogeneity in SARS-CoV-2-specific memory immune responses following infection. There remains a critical need to identify host immune biomarkers predictive of clinical and immunological outcomes in SARS-CoV-2-infected patients.

**Methods:** Leveraging longitudinal samples and data from a clinical trial (N=108) in SARS-CoV-2-infected outpatients, we used host proteomics and transcriptomics to characterize the trajectory of the immune response in COVID-19 patients. We characterized the association between early immune markers and subsequent disease progression, control of viral shedding, and SARS-CoV-2-specific T cell and antibody responses measured up to 7 months after enrollment. We further compared associations between early immune markers and subsequent T cell and antibody responses following natural infection with those following mRNA vaccination. We developed

machine-learning models to predict patient outcomes and validated the predictive model using data from 54 individuals enrolled in an independent clinical trial.

**Results:** We identify early immune signatures, including plasma RIG-I levels, early IFN signaling, and related cytokines (CXCL10, MCP1, MCP-2, and MCP-3) associated with subsequent disease progression, control of viral shedding, and the SARS-CoV-2-specific T cell and antibody response measured up to 7 months after enrollment. We found that several biomarkers for immunological outcomes are shared between individuals receiving BNT162b2 (Pfizer–BioNTech) vaccine and COVID-19 patients. Finally, we demonstrate that machine-learning models using 2–7 plasma protein markers measured early within the course of infection are able to accurately predict disease progression, T cell memory, and the antibody response post-infection in a second, independent dataset.

**Conclusions:** Early immune signatures following infection can accurately predict clinical and immunological outcomes in outpatients with COVID-19 using validated machine-learning models.

**Funding:** Support for the study was provided from National Institute of Health/National Institute of Allergy and Infectious Diseases (NIH/NIAID) (U01 AI150741-01S1 and T32-AI052073), the Stanford's Innovative Medicines Accelerator, National Institutes of Health/National Institute on Drug Abuse (NIH/NIDA) DP1DA046089, and anonymous donors to Stanford University. Peginterferon lambda provided by Eiger BioPharmaceuticals.

## Editor's evaluation

This manuscript uses a multi-omics approach to investigate how early immune markers in blood predict subsequent clinical outcome and immune responses. The study uses samples from a previous trial and identifies several immune markers associated with later clinical and immunological outcomes in this cohort. An important next step will be to validate this in other cohorts and test the utility of this in a clinical setting.

## Introduction

The great majority of severe acute respiratory syndrome-related coronavirus 2 (SARS-CoV-2) infections initially present with mild to moderate symptoms. However, some patients with initially mild infections progress to more severe disease requiring hospitalization and/or prolonged symptoms leading to sustained disability (*Wu and McGoogan, 2020*). Early identification of these patients would help guide treatment decisions, including the use of monoclonal antibodies and novel antivirals that can prevent disease progression. Moreover, mild infections are an important contributor to ongoing viral transmission, and there is substantial heterogeneity in the degree of the SARS-CoV-2-specific memory immune response following infection (*Dan et al., 2021*; *Wheatley et al., 2021*; *Zuo et al., 2021*). An improved understanding of host determinants of clinical, virological, and immunological outcomes of SARS-CoV-2 infection can help spur the development of novel therapeutic and vaccination strategies.

The early host response to acute SARS-CoV-2 infection likely plays a critical role in determining disease outcome and generation of virus-specific memory immune responses. Nucleic acid pattern recognition receptors (PRRs) mediate the early detection and host response to viral infections, with RNA virus recognition thought to occur mainly in the endosomal and/or cytosolic compartment by two different PRRs: toll-like receptors (TLRs) and RIG-I-like receptors (RLRs). Viral recognition by TLRs and RLRs typically triggers a signaling cascade leading to induction of pro-inflammatory cytokines and type I and type III IFNs, which provide both a cell-intrinsic state of viral resistance and help coordinate the generation of adaptive immune responses (*Park and Iwasaki, 2020*). Most studies evaluating the early host response have been cross-sectional and/or performed in patients already with severe disease (*Arunachalam et al., 2020*; *Wilk et al., 2020*; *Abers et al., 2021*; *Liu et al., 2021*; *Lucas et al., 2020*; *Schulte-Schrepping et al., 2020*; *Wilk et al., 2021*; *Stephenson et al., 2021*); longitudinal studies among those presenting earlier in disease, with prospective clinical outcomes, are lacking (*Talla et al., 2021*).

Following initial infection, SARS-CoV-2-specific memory immune responses result in protection from reinfection, likely mediated in part by SARS-CoV-2-specific memory T cell and both binding and neutralizing antibody responses (*Rodda et al., 2021*; *Wu et al., 2021*; *Crawford et al., 2021*; *Chakraborty et al., 2022*). However, there is considerable heterogeneity in the T cell and antibody

response following natural SARS-CoV-2 infection, with rapid decay in early convalesce, providing valuable opportunities to identify key immune components that are associated with the establishment and durability of memory immune responses. Although mRNA vaccination also leads to establishment of protective immunological memory (*Skowronski and De Serres, 2021*), this memory wanes (*Goldberg et al., 2021*). Comparing responses induced by vaccination with those induced by natural SARS-CoV-2 infection could potentially guide researchers to better understand determinants of durable protective immunity and improve vaccine design.

In this paper, we utilized a multi-omics approach to define early infection signatures following SARS-CoV-2 infection that predict subsequent disease progression, oropharyngeal viral load, and SARS-CoV-2-specific memory immune responses. We leveraged longitudinal samples collected from outpatients enrolled in a randomized controlled trial of a type III IFN, Peginterferon lambda-1a (Lambda, NCT04331899) (*Jagannathan et al., 2021*). In this trial, outpatients with initially mild to moderate COVID-19 were recruited within 72 hr of diagnosis and followed through 7 months post-infection. We observed sequential activation of immune modules in initially mild to moderate COVID-19 patients within the first 2 weeks of symptom onset, including IFN responses, T cell activation, and B cell responses. We identified variations in plasma proteins, early IFN signaling, and downstream cytokines (MCP1, MCP-2, and MCP-3) that were associated with multiple patient outcomes, including disease progression, viral load, memory T cell activity, and S protein-binding IgG levels measured up to 7 months after enrollment. We also compared the immune response in COVID-19 patients to the response following COVID-19 mRNA vaccination and identified biomarkers for immunological outcomes, including CXCL10, MCP-1, and IFN gamma, that are shared between individuals receiving BNT162b2 (Pfizer–BioNTech) vaccine and COVID-19 patients. Finally, we demonstrate that a machine-learning model using 2–7 plasma protein markers is able to accurately predict disease progression and the magnitude of the SARS-CoV-2-specific CD4+ T cell response and antibody response in a second, independent cohort.

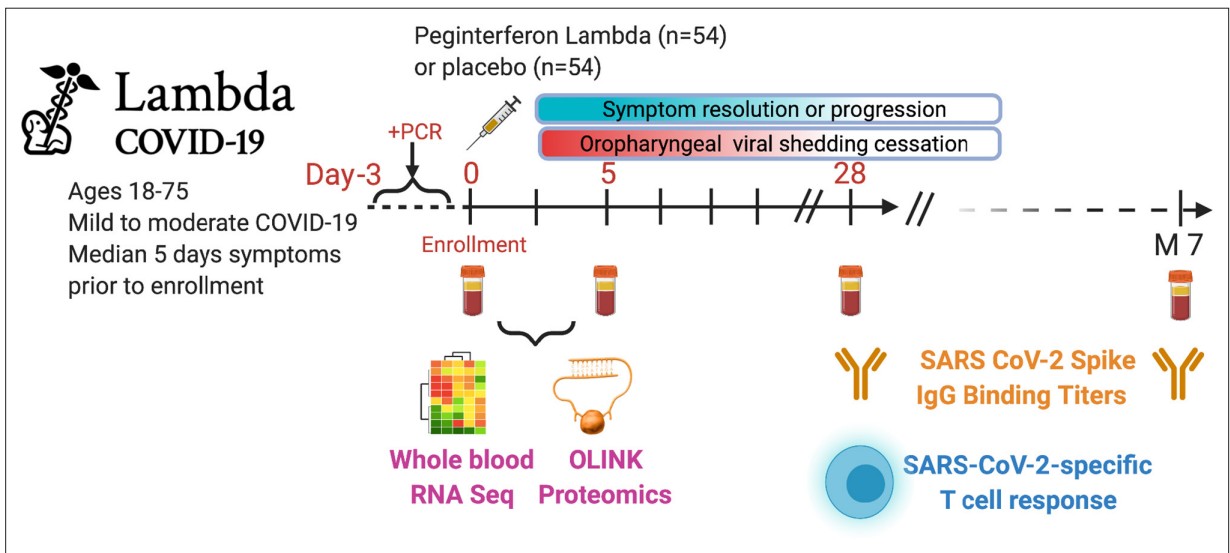

**Figure 1.** Study schema. Outpatients (n=108) with PCR-confirmed severe acute respiratory syndrome-related coronavirus 2 (SARS-CoV-2) infection and swab obtained within 72 hr of randomization were enrolled in a phase 2 clinical trial of subcutaneous Peginterferon lambda vs. placebo. In-person follow-up visits were conducted at days 1, 3, 5, 7, 10, 14, 21, 28, and month 7 post-enrollment, with assessment of symptoms and vitals and collection of oropharyngeal swabs for SARS-CoV-2 testing. Blood obtained at days 0 and 5 were evaluated by whole blood transcriptomics (RNA sequencing), plasma proteomics (Olink), and SARS-CoV-2-specific antibodies. Clinical outcomes assessed included duration of symptoms and duration of virological shedding. Immunological outcomes assessed including SARS-CoV-2-specific T cell responses at day 28, and antibody responses at day 28 and month 7. Created with biorender.com.

## Results

### Transcriptomic and proteomic profiles correlate with the time to symptom onset in COVID-19 patients

We recruited 108 participants with initially asymptomatic to moderate COVID-19 at diagnosis into this study. The median age of participants was 37 years (range 18–71) with 57% male and 62% of Latinx ethnicity (*Jagannathan et al., 2021*). Eight (6.7%) participants were asymptomatic at baseline. Of those with symptoms, the median duration of symptoms prior to randomization was 5 days (IQR 3–6 days *Supplementary file 1*).

Subjects were randomized to receive a single dose of Peginterferon lambda or placebo at their first visit and followed up to 7 months post-enrollment (*Figure 1*). The median duration of viral shedding post-enrollment was 7 days, and symptoms were 8 days, and this did not differ between participants randomized to Peginterferon lambda compared with placebo (*Jagannathan et al., 2021*). To profile the immune response in these patients, we conducted whole blood RNA-sequencing and plasma protein profiling with multiplex Olink panels (inflammation and immune response panels, n=184 proteins) using blood samples collected at day 0 and day 5 after enrollment. We assessed SARS-CoV-2-specific CD4+ T cell responses by intracellular cytokine staining using peripheral blood mononuclear cells (PBMC) collected at day 28 after enrollment. We also measured IgG-binding titers against the SARS-CoV-2 full-length spike protein (S) using plasma collected at day 0, day 5, day 28, and month 7 (*Figure 1*).

We first examined antibody levels and transcriptomic profiles at day 0 and day 5 after enrollment in both patients randomized to Peginterferon lambda and placebo. Based on the subject-reported symptom starting date, samples at day 0 were collected a median 5 days after symptom onset (range –1 to 15; *Figure 2A*). As expected, we observed a positive correlation between the S protein-binding IgG levels at enrollment and the time since symptom onset (*Figure 2B*). We performed principal component analysis (PCA) of transcriptomic data and calculated the correlation between the first two principal components (PCs) and other clinical variables. We found that PC1 had the strongest association with the time since symptom onset and the IgG titer, suggesting that whole blood transcriptomic profiles capture the progression of the immune response in COVID-19 patients (*Figure 2C–E*). We also performed PCA on the Olink data. Similar to results from the analysis of transcriptomics data, Olink data were associated with disease progression, as indicated by the high correlation between PC2 and the time since symptom onset (*Figure 2F–H*). We also observed an association between PC1 and age, which captures the impact of age on the plasma protein landscape in COVID-19 patients.

We previously reported that Peginterferon lambda treatment neither shortened the duration of SARS-CoV-2 viral shedding nor improved symptoms in outpatients with COVID-19 in this study (*Jagannathan et al., 2021*). However, a similar, smaller study of 60 participants found that Peginterferon lambda was associated with more rapid viral clearance 7 days post-treatment among a subset of individuals with high baseline viral loads (*Feld et al., 2021*). We therefore assessed whether subcutaneous Peginterferon lambda treatment altered immune phenotypes post-treatment. PCA revealed that transcriptional and proteomics profiles at day 5 post-treatment were similar between Peginterferon lambda and placebo treatment arms (*Figure 2D and G*, and *Figure 2—figure supplement 1*). We also tested the effect of Peginterferon lambda treatment on individual immune measures at day 5 post-treatment. There were no significant differences in blood transcription modules (BTMs) between groups, and only two plasma proteins (HSD1B1 and LAMP3) were significantly affected by Peginterferon lambda treatment (*Figure 2I*). Furthermore, we found no significant differences in SARS-CoV-2-specific T cell responses (at day 28 after enrollment) and antibody responses (at day 28 and month 7 after enrollment) between the two treatment arms (*Figure 2—figure supplement 1*), as reported previously (*Chakraborty et al., 2021*; *van der Ploeg et al., 2022*). Taken together, Peginterferon lambda treatment did not show substantial effects on the immune response in COVID-19 outpatients.

### Trajectory analysis reveal sequential activation of immune modules in COVID-19 patients

We next characterized the trajectory of early transcriptomic and proteomic responses using the RNA-seq and Olink data as a function of time since symptom onset. To reduce the dimensionality and improve interpretability, we calculated the enrichment score of different immune modules (based

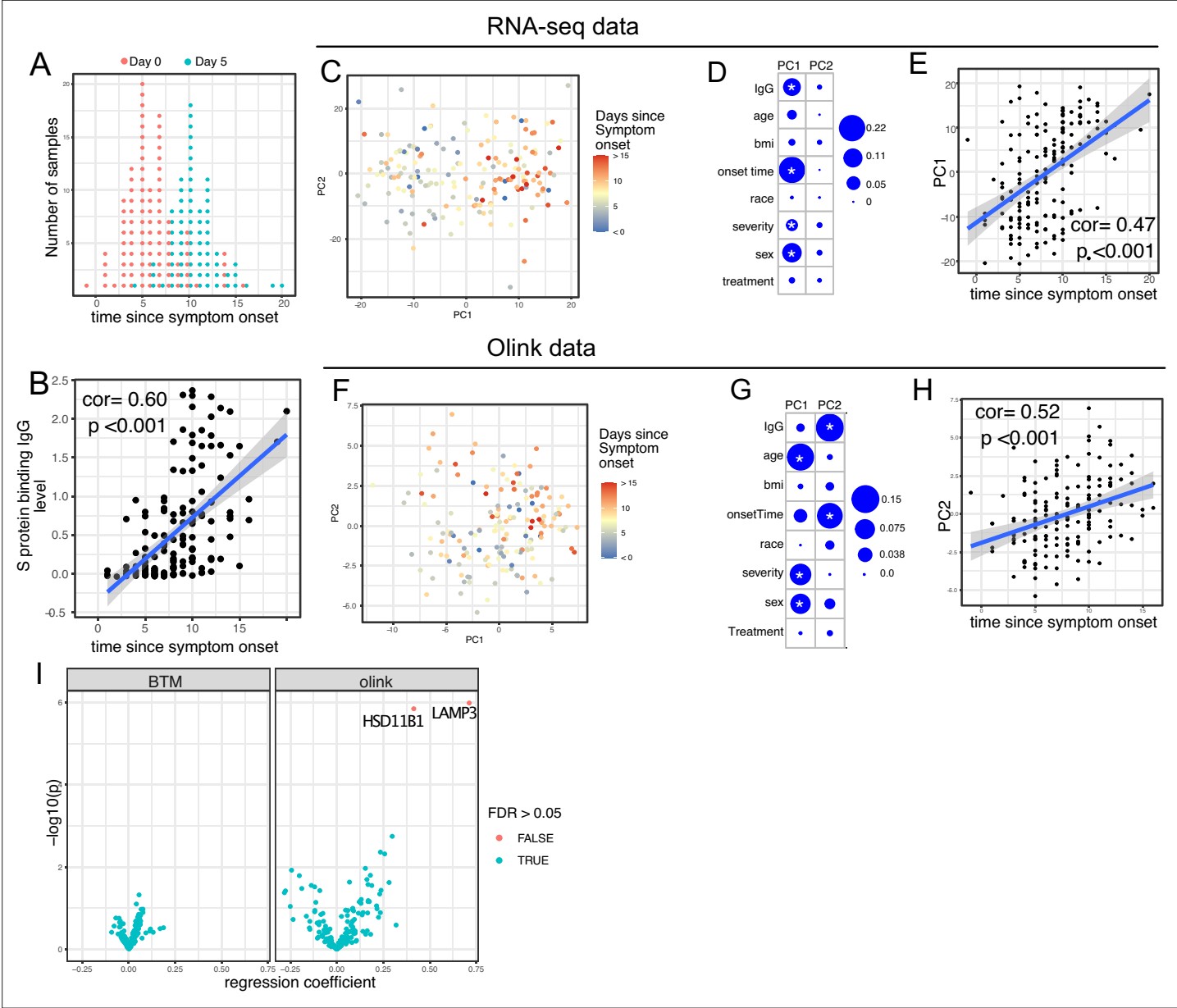

**Figure 2.** Transcriptomics and proteomics profiles correlate with the time to symptom onset in COVID-19 patients. (**A**) The distribution of RNA-seq sample collection time in respect to symptom onset. The colors of the dots represent the sample collection time from the enrollment. Asymptomatic cases are not shown. (**B**) Scatter plot showing the positive correlation between severe acute respiratory syndrome-related coronavirus 2 (SARS-CoV-2) spike (S) protein-binding IgG antibody level and the time since symptom onset. Pearson correlation is reported. (**C**) Principal component analysis (PCA) plot of the RNA-seq samples. The colors of the dots represent the sample collection time from the enrollment. (**D**) The percent of the variances of PC1 and PC2 explained by different clinical variables. Stars indicate false discovery rate (FDR) <0.05. (**E**) Scatter plot showing the positive correlation between PC1 of the RNA-seq data and the time since symptom onset. Pearson correlation is reported. (**F**) PCA plot of the Olink proteomics data. The colors of the dots represent the sample collection time from the enrollment. (**G**) The percent of the variances of PC1 and PC2 explained by different clinical variables. Stars indicate FDR <0.05. (**H**) Scatter plot showing the positive correlation between PC2 of the Olink data and the time since symptom onset. Pearson correlation is reported. (**I**) Volcano plot showing the effect (measured as regression coefficient) and p value of Peginterferon lambda treatment on blood transcription modules and plasma proteins at day 5 post-treatment.

The online version of this article includes the following figure supplement(s) for figure 2:

**Figure supplement 1.** Immune profiles comparing Peginterferon Lambda vs. Placebo.

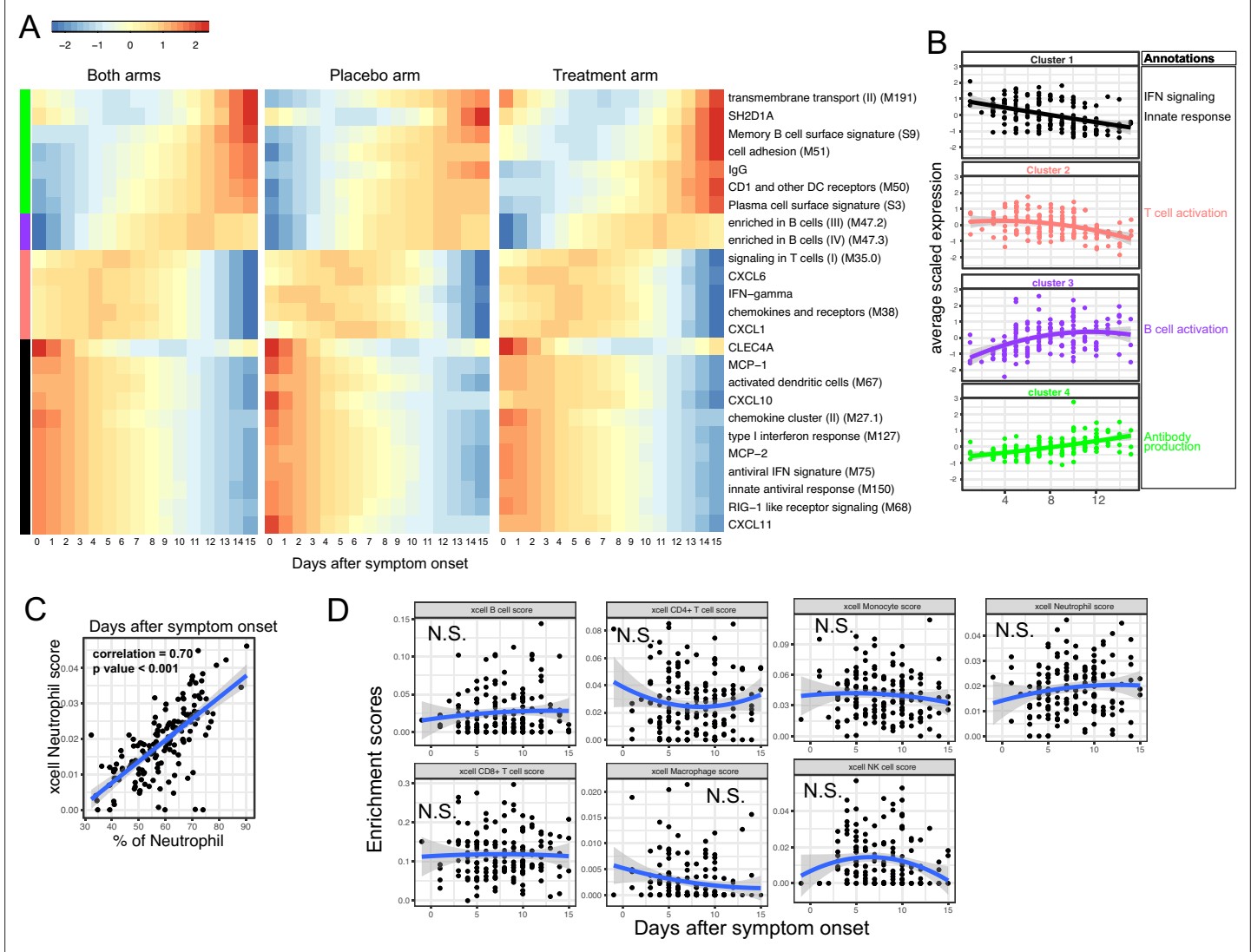

**Figure 3.** Trajectory analysis reveals sequential activation of immune modules in COVID-19 patients. (**A**) The fitted expression level of immune modules and plasma proteins at 0–15 days after symptom onset. The values are calculated by fitting quadratic regressions and are scaled to a mean of 0 and an SD of 1. The left, middle, and right panels show the fitted trajectory using data from both arms, placebo arm only, and Peginteferon lambda arm only. Modules and proteins with false discovery rate (FDR) <0.05 (based on data from both arms) are shown. The color bar on the left side shows the clustering membership of the modules and plasma proteins. (**B**) The average trajectory of the clusters. We scaled the expression level of each module and plasma proteins to a mean of 0 and an SD of 1. We then calculated the average-scaled expression of all members in the clusters. Each dot represents the mean expression in each blood sample. The lines represent the fitted quadratic regression. The gray areas represent the 95% CI. (**C**) We estimated the Spearman correlation between the neutrophil enrichment score using the xCell. The plot shows the correlation between the xCell score and the counted neutrophil percentage in whole blood. (**D**) The relationship between xCell enrichment score and days after symptom onset.

The online version of this article includes the following figure supplement(s) for figure 3:

**Figure supplement 1.** Gene Ontology analysis of immune profiles in COVID-19 outpatients.

on BTM [*Li et al., 2014*]) from the RNA-seq data. We then combined the enrichment scores and Olink measurements into a single dataset for the trajectory analysis. We fitted the data with quadratic regression to capture the non-linear dynamics of the modules and proteins. In addition, 'treatment' was included as a variable to control for differences between the two arms. We identified 15 immune modules and 10 plasma proteins that varied as a function of time since symptom onset (false discovery rate [FDR] <0.05; *Figure 3A* and *Supplementary file 2*). Among them, 16 immune modules or proteins showed non-linear dynamics, as indicated by significant coefficients of the quadratic term (*Supplementary file 2*).

We performed clustering analysis and identified four clusters based on the trajectory of the significant modules and proteins (*Figure 3A–B*). Cluster 1 contains IFN-related modules and proteins known to be activated by IFN signaling, including MCP-1, MCP-2, CXCL10, and CXCL11 (*Lehmann et al., 2016*; *Lee et al., 2009*; *Moll et al., 2011*). The trajectories in cluster 1 already reached the peak at the time of symptom onset and monotonically decreased over time. The trajectories in cluster 2 peaked at 1–5 days after symptom onset and contain IFN-γ and modules related to T cell activation. Interestingly, it also contains several myeloid cell attracting chemokines (CXCL1 and CXCL6) and the innate cell response modules. Cluster 3 peaked between 10 and 14 days after the symptom onset and is characterized by modules related to B cells. Cluster 4 trajectories monotonically increase after symptom onset and are characterized by the increasing S protein-binding IgG level and related plasma B cell modules. The trajectory analysis revealed the sequential activation of IFN signaling, myeloid cells, IFN-γ, T cells, B cell, and antibody production within the first 15 days of symptom onset. Consistent with the BTM analysis, a pathway analysis using Gene Ontology identified a similar sequential activation of immune pathways within the first 15 days of symptom onset (*Figure 3—figure supplement 1A*). The immune trajectories of the two individual treatment arms were similar to the trajectories of the combined dataset (*Figure 3A*), suggesting that Peginterferon lambda treatment does not significantly affect early immune trajectories after COVID-19 infection.

To characterize how the composition of blood immune cells change over time, we used a previously established tool named xCell to estimate the enrichment score of the major immune cells (*Aran et al., 2017*). As a positive control, we compared the neutrophil score with the neutrophil count data obtained from clinical lab tests and found high correlation between them (*Figure 3C*). Quadratic regression did not find significant associations between the major cell types and the time since symptom onset (*Figure 3D*). The results suggest that the trajectories of different immune modules (*Figure 3A*) are mainly driven by the activation of corresponding immune cells rather than the composition change of major immune cell types.

## Variations in early immune responses are associated with disease severity in COVID-19 patients

We next sought to identify immune modules and plasma proteins associated with disease progression in COVID-19 outpatients. At the time of enrollment (day 0), the majority of subjects showed either mild to moderate symptoms that subsequently resolved (n=92) or were asymptomatic (n=8). However, eight patients with initially mild to moderate symptoms later developed progressive and more severe symptoms and presented to the emergency department or were hospitalized (median 2 days to progression, range 1–13 days; *Supplementary file 1*). We defined these individuals as 'progressors' and used regression models to identify immune modules and plasma proteins to compare these participants with those who did not seek care at the hospital (non-progressors), while controlling for days after symptom onset and treatment arm.

As two positive controls, we confirmed well-documented findings that lymphocyte percentages were negatively correlated with symptom severity, and neutrophil percentages were positively correlated with symptom severity (*Figure 4A*; *Goyal et al., 2020*). In addition, our regression analysis identified four immune modules and 25 plasma proteins that were significantly associated with disease progression (FDR <0.05, *Figure 4B–C*, and *Supplementary file 3*). The proteins and modules from cluster 1 (as identified above in *Figure 3A*), including the type I IFN response, RIG-I signaling, and multiple proteins known to be induced by IFN signaling including CXCL10 (also known as IFN gamma-induced protein, IP-10), MCP-1, MCP-2, and CXCL11, were significantly enriched in individuals who experienced disease progression vs. non-progressors (Fisher's exact test, p<0.001). As our regression analysis excluded asymptomatic individuals, we performed one-way ANOVA analysis without adjusting for symptom onset time. The results from the ANOVA analysis were consistent with the regression analysis (*Figure 4D*). Pathway analysis using Gene Ontology identified similar IFN and RIG-I-related pathways to be associated with the symptom progression (*Figure 3—figure supplement 1B*). Together, these data highlight an association between early innate immune activation and disease progression.

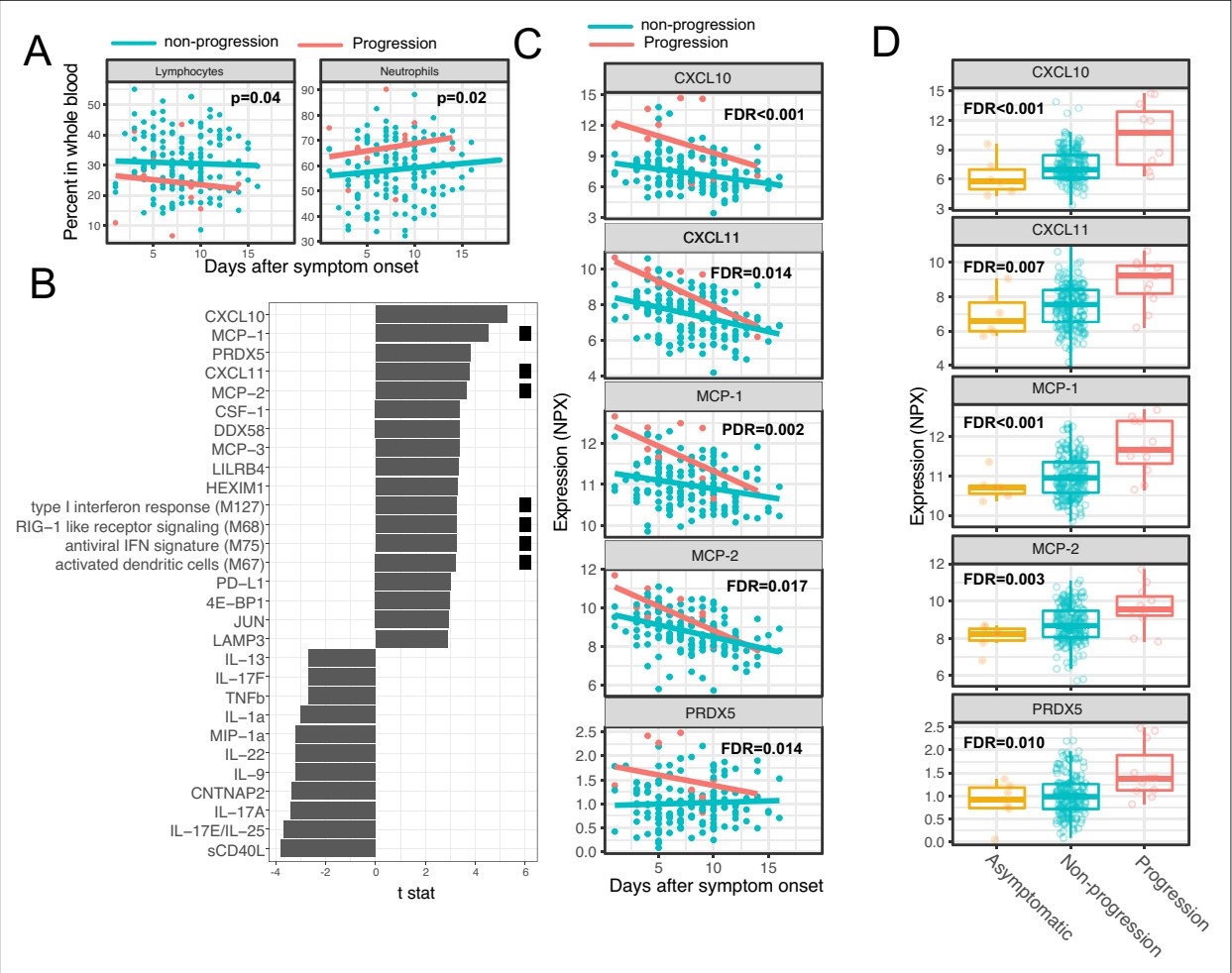

**Figure 4.** Variations in early immune responses are associated with disease severity in COVID-19 patients. (**A**) Scatter plot comparing the percentage of lymphocytes and neutrophils in whole blood between moderate and severe cases. The lines represent the fitted linear relationship between the percentages and the time after symptom onset. We fitted regression models to test the relationship between the immune measurements and disease progression while controlling for the time after symptom onset and treatment. The p values for the disease progression are reported. (**B**) We fitted regression models to test the relationship between the immune measurements and symptom severity while controlling for the time after symptom onset. The bar plot shows the t score of the regression coefficient for disease progression. The colored squares represent the clusters each immune measurement belongs to. The clusters are defined in **Figure 3A**. False discovery rate (FDR) represents the p value of the regression coefficient of the disease progression term after multiple testing adjustment. (**C**) Scatter plot comparing the plasma protein levels between moderate and severe cases. The lines represent the fitted linear relationship between the percentages and the time after symptom onset. The top five significant proteins are shown. Data from asymptomatic cases are omitted, as their symptom onset time was unknown. FDR are the same as in B. (**D**) Box plots comparing the plasma protein levels between asymptomatic, non-progressed, and progressed cases. FDR represent the p value from one-way ANOVA after multiple testing adjustment.

## Early proteomic and transcriptomic signatures show long-term association with virological and immunological outcomes

We next examined associations between plasma proteins measured early in the course of infection and oropharyngeal viral load (measured by the area under the Ct curve from day 0 to 14 post-enrollment), SARS-CoV-2-specific T cells measured 28 days post-enrollment (***Figure 5—figure supplement 1***; ***Supplementary file 4***), and anti-S-binding antibodies measured at 28 days and 7 months post-enrollment, while controlling for days since symptom onset and treatment arm. Importantly, these virological and immunological outcomes were not significantly associated with disease progression, suggesting that they are not driven by more severe disease (***Figure 5—figure supplement 2***). We identified 38 plasma proteins significantly associated with oropharyngeal viral load (top 10 significant proteins shown in ***Figure 5A***, ***Supplementary file 3***). Higher levels of several of these proteins were

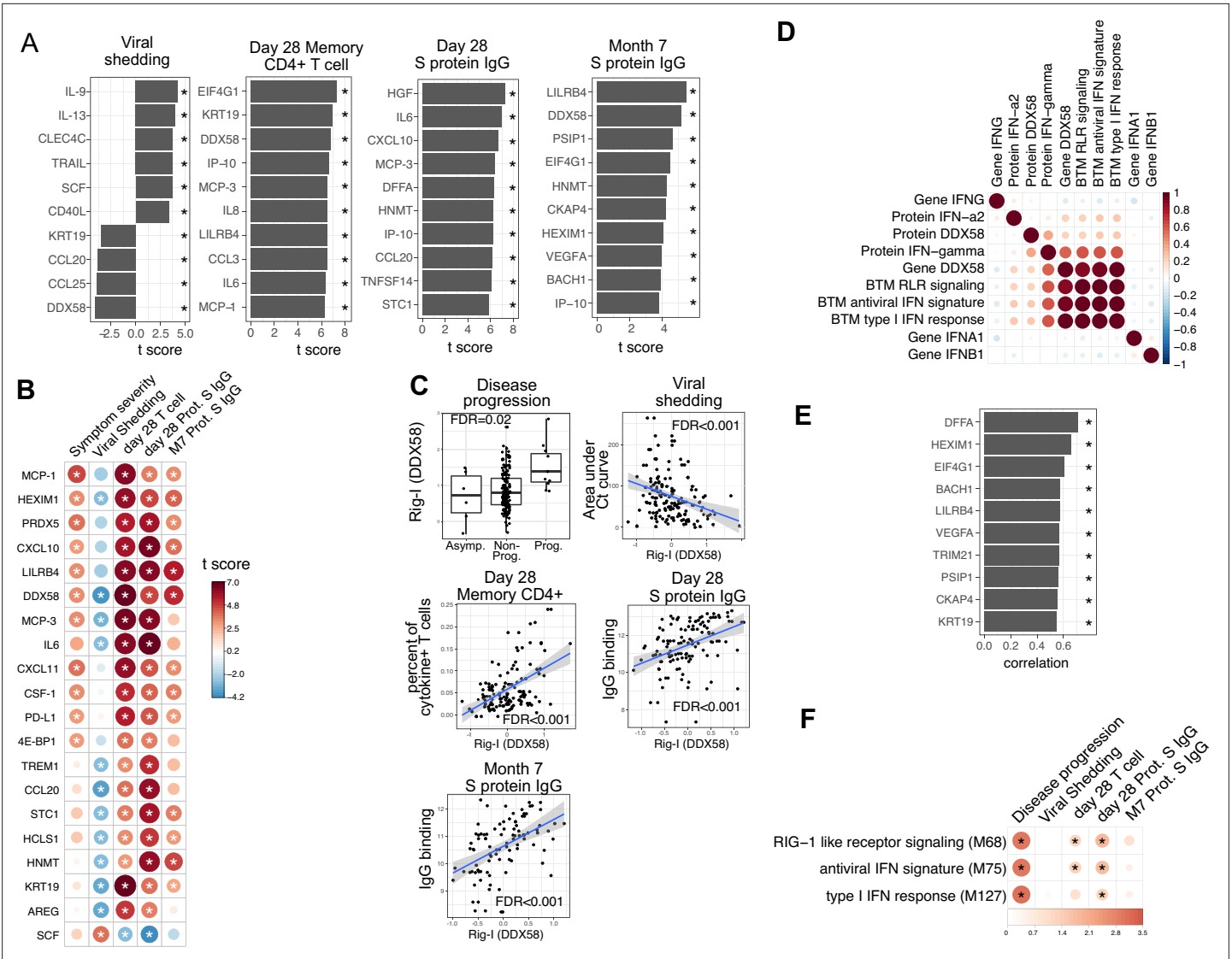

**Figure 5.** Plasma RIG-I is a biomarker for disease progression, viral shedding, T cell activity, and spike (S)-binding IgG levels. (**A**) The association between plasma proteins and viral shedding, memory T cell activity, and anti-S-binding IgG levels. Memory CD4+ T cell activities are measured by the percent of cytokine positive T cells (TNF-α+ or TNFγ+ or IL21+) after S protein stimulation. T cells are collected from patients 28 days after enrollment. S protein-binding IgG levels are measured 28 days or 7 months after enrollment. We fitted regression models to test the relationship between the immune measurements and viral shedding, memory T cell activity, and S protein-binding IgG level while controlling for the time after symptom onset and treatment. The bar plot shows the t score of the regression coefficient for viral shedding, memory T cell activity, and anti-S-binding IgG levels. (**B**) Association between plasma proteins and multiple outcomes. The heatmaps include immune measurements that are significantly associated (indicated by stars) with at least three outcomes. (**C**) Correlation between plasma RIG-I (DDX58) and disease progression, viral shedding, memory T cell activity, and S protein-binding IgG levels. (**D**) Correlation between plasma RIG-I protein and selected level of plasma proteins, genes, and blood transcription modules (BTMs). (**E**) The top 10 plasma proteins correlated with plasma RIG-I protein. (**F**) The association between RIG-I and IFN related BTMs and the outcomes of COVID-19 patients. Stars represent false discovery rate (FDR) <0.05.

The online version of this article includes the following figure supplement(s) for figure 5:

**Figure supplement 1.** Measurement of severe acute respiratory syndrome-related coronavirus 2 (SARS-CoV-2) specific T cell responses by intracellular cytokine staining.

**Figure supplement 2.** Boxplots visualizing the relationship between disease progression and other COVID-19 outcomes (viral shedding, memory T cell activity, and anti-spike-binding IgG levels).

**Figure supplement 3.** We estimated the association between immune measures (blood transcription module [BTM] and Olink proteins measures) and COVID-19 outcomes using data from both treatment arms and from only placebo arm of the lambda trial.

**Figure supplement 4.** We performed t tests to compare the change of the 20 most relevant (as listed in **Figure 5B**) plasma proteins between day 0 and day 5 of the study.

inversely correlated with viral load, including the cytosolic RNA sensor RIG-I (gene symbol DDX58), chemokines (CCL20 and CCL25), and other proteins (Keratin 19 [KRT19], amphiregulin [AREG]) previously shown to be upregulated in COVID-19 patients (*Talla et al., 2021*). In contrast, higher levels of C-type lectin domain family 4 member C (CLEC4C, expressed by plasmacytoid dendritic cells) and TNF-related apoptosis-inducing ligand (TRAIL) were associated with higher oropharyngeal viral loads.

We identified 86 plasma proteins that were significantly associated with SARS-CoV-2-specific T cell responses at day 28, and 83 and 55 plasma proteins significantly associated with S protein-binding IgG at day 28 and month 7, respectively (top 10 significant proteins shown in *Figure 5A*, *Supplementary file 3*). Several proteins were associated with higher levels of SARS-CoV-2-specific T cells and the antibody response, including RIG-I (gene symbol DDX58), chemokines (CXCL11), and other proteins (KRT19 and AREG) also associated with control of viral load (*Figure 5A and B*). Notably, regression analysis demonstrated that neither HSD1B1 nor LAMP3 – the two proteins that were influenced by Peginterferon lambda at Day 5 (*Figure 2I*) – were associated with patient outcomes, including disease progression, oropharyngeal viral load, SARS-CoV-2 specific T cell responses, or antibody responses measured at day 28 (*Supplementary file 3*).

Altogether, we identified 20 plasma proteins that were correlated with three out of four patient outcomes (*Figure 5B*). To ensure that there were not significant interactions between these early plasma proteins, treatment assignment, and patient outcomes, we estimated associations using data from the placebo arm of the Peginterferon lambda trial only, with similar results to the pooled analysis (*Figure 5—figure supplement 3*). Furthermore, Peginterferon lambda treatment did not significantly affect the change of these 20 proteins between day 0 and day 5 (*Figure 5—figure supplement 4*). Interestingly, higher plasma levels of RIG-I (gene symbol DDX58) were significantly associated with all examined clinical, virological, and immunological outcomes (*Figure 5C*), including disease progression, lower oropharyngeal viral load, increased SARS-CoV-2-specific T cell responses, and increased levels of S protein-binding IgG to SARS-CoV-2. Since RIG-I is a cytosolic PRR that, upon recognition of short viral dsRNA during a viral infection, leads to upregulation of IFN signaling (*Matsumiya and Stafforini, 2010*), we explored associations between plasma RIG-I levels and related immune measurements, including the mRNA-level and protein-level expression of RIG-I and IFNs, as well as RIG-I and IFN-related modules. We found that the plasma RIG-I levels were only modestly correlated with mRNA-level expression of RIG-I (correlation = 0.23, p value=0.004, *Figure 5D*), as well as RIG-I signaling and IFN-related modules (*Figure 5D*). In contrast, we found a strong correlation between plasma levels of RIG-I and plasma levels of DNA Fragmentation Factor Subunit Alpha (DFFA), an intracellular protein known to be involved in apoptosis (*Figure 5H*; *Thomas et al., 2000*; *Zhang and Xu, 2000*). In addition, many of the other plasma proteins correlated with plasma RIG-I levels were intracellular proteins (*Figure 5E*), suggesting that plasma RIG-I levels may be driven by both increased intracellular RIG-I expression, as well as a cell death process that releases intracellular protein into the plasma. Analysis of RNA-sequencing data also identified associations between RIG-I, IFN signaling, and cell death-related modules and clinical and immunological outcomes of patients (*Figure 5F* and *Figure 3—figure supplement 1C*). As plasma levels of RIG-I were not significantly associated with time since symptom onset (*Supplementary file 2*), these data suggest that plasma RIG-I levels might serve as a powerful and stable biomarker for predicting several clinical, virological, and immunological outcomes in patients with COVID-19.

## Similar trajectories of immune responses induced by SARS-CoV-2 infection and COVID-19 mRNA vaccine

The BNT162b2 (Pfizer–BioNTech) vaccine has been widely used throughout the world and is highly effective in preventing SARS-CoV-2 infection, as well as protecting patients from severe symptoms after infections (*Skowronski and De Serres, 2021*). We leveraged a recently published dataset from a BNT162b2 vaccine study to compare the immune response induced by COVID-19 vaccine and SARS-CoV-2 infections (*Arunachalam et al., 2021*). Comparison of the datasets reveals that the immune response after the first dose of vaccination (day 0 to day 21) largely mirrors the trajectory of immune response after SARS-CoV-2 infection. Early proteins and BTMs in the SARS-CoV-2 infection dataset, including IFNγ, MCP1, CXCL11, MCP2, CXCL10, and IFN-related transcriptional modules, are upregulated within the first 7 days of the vaccination. Late immune markers in the SARS-CoV-2 infection dataset, including SLAMF1, TNFRSF9, CCL3, CCL4, TGFα, TNFSF14, and B cell-related transcriptional

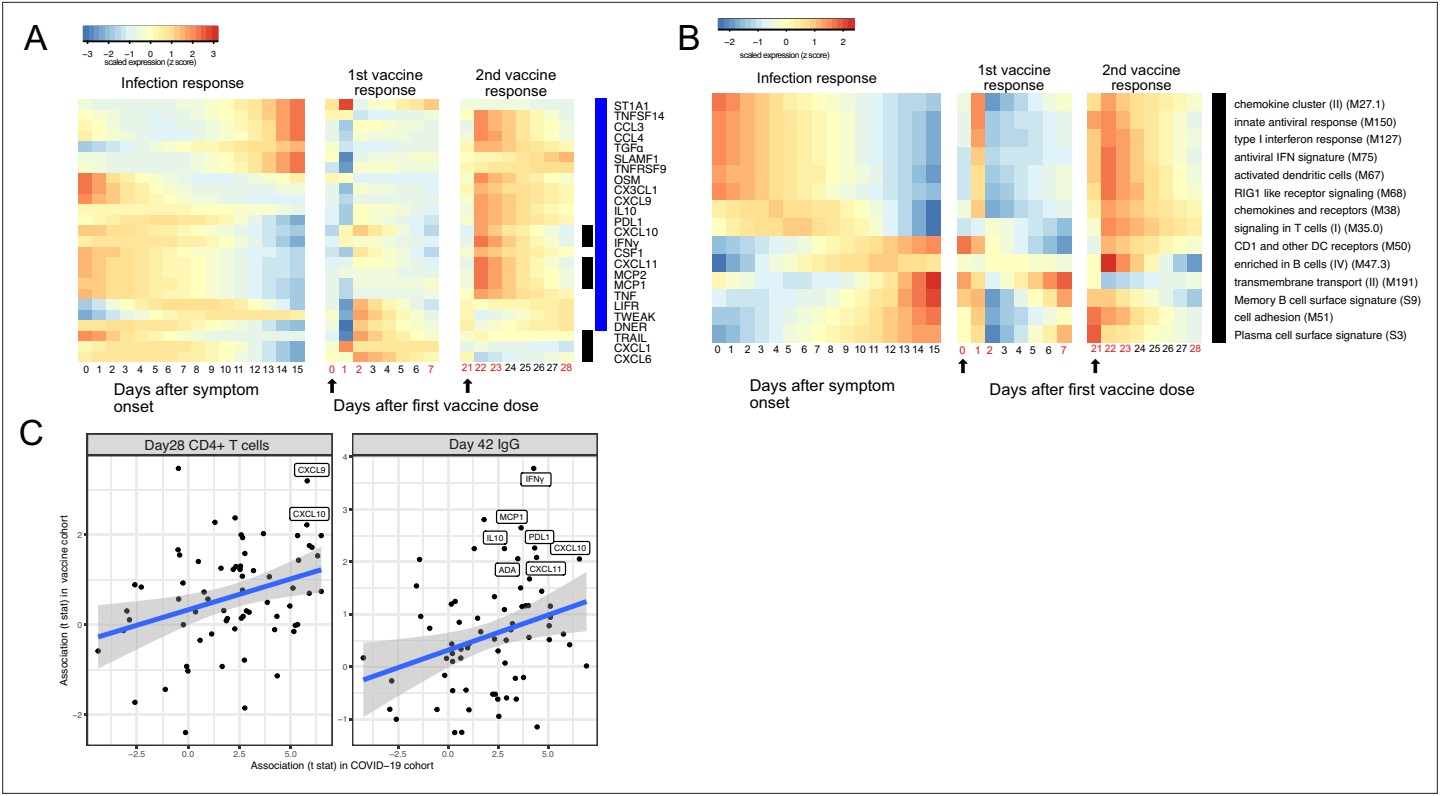

**Figure 6.** Comparing the immune response induced by severe acute respiratory syndrome-related coronavirus 2 (SARS-CoV-2) infection and COVID-19 vaccine (BNT162b2). (**A–B**) The heat map shows the expression level of plasma proteins (**A**) and blood transcription modules (BTMs) (**B**) at 0–15 days after symptom onset in COVID-19 patients (left), 0–7 days after the first dose of vaccination, and 21–28 days after the second dose of vaccination in healthy individuals (right). The values from the SARS-CoV-2 dataset are calculated by fitting quadratic regressions and are scaled to a mean of 0 and an SD of 1. The values from the vaccination dataset are computed by fitting using linear interpolation between the measured time points (days in red color) and are scaled to a mean of 0 and an SD of 1. The black bar on the right side shows the protein markers that are significantly associated with time in COVID-19 patients (false discovery rate [FDR] <0.05). The blue bar on the right side shows the protein markers that are significantly associated with time after vaccination (FDR <0.05). The black arrows indicate the time of the first and the second doses of vaccination. (**C**) Comparing the biomarkers of immune outcomes in COVID-19 dataset and the BNT162b2 dataset. We fitted regression models to test the association between the highest level of protein markers after first vaccination (day 0 and day 21) and the T cell (left) and antibody responses (right). The t statistics from the regression model is compared with the t statistics from the COVID-19 dataset.

modules, are upregulated much later and show highest levels 21 days after the vaccination (*Figure 6A and B*). In contrast, the response after the second dose of vaccine (day 22 to day 28) is characterized by fast upregulation of both early and late immune measurements (*Figure 6A–B*). Interestingly, three proteins that are significantly upregulated in SARS-CoV-2 patients were not induced after the second dose of vaccine, including TRAIL, CXCL1, and CXCL6. All three proteins are highly expressed in neutrophils (*Uhlén et al., 2015*) and have been shown to regulate neutrophil recruitment (CXCL1 and CXCL6) or apoptosis (TRAIL) during inflammation (*Jovic et al., 2016*; *Sawant et al., 2016*; *Renshaw et al., 2003*). The lack of TRAIL, CXCL1, and CXCL6 suggests an absence of neutrophil response to the second dose of vaccine.

We next examined associations between plasma proteins and immunological outcomes of BNT162b2 vaccination. The associations identified in the vaccine dataset are largely consistent with the associations identified in the COVID-19 infection dataset (*Figure 6C*). In particular, proteins that are associated with the T cell (CXCL9 and CXCL10) and antibody responses (INF-γ, MCP1, L10, PDL1, CXCL10, ADA, and CXCL11) after infection were also associated with the T cell and antibody responses after vaccination, highlighting important similarities between infection- and vaccine-induced immunity. Furthermore, these results suggest that these plasma biomarkers may be useful correlates of protective immunity following both natural infection and vaccination.

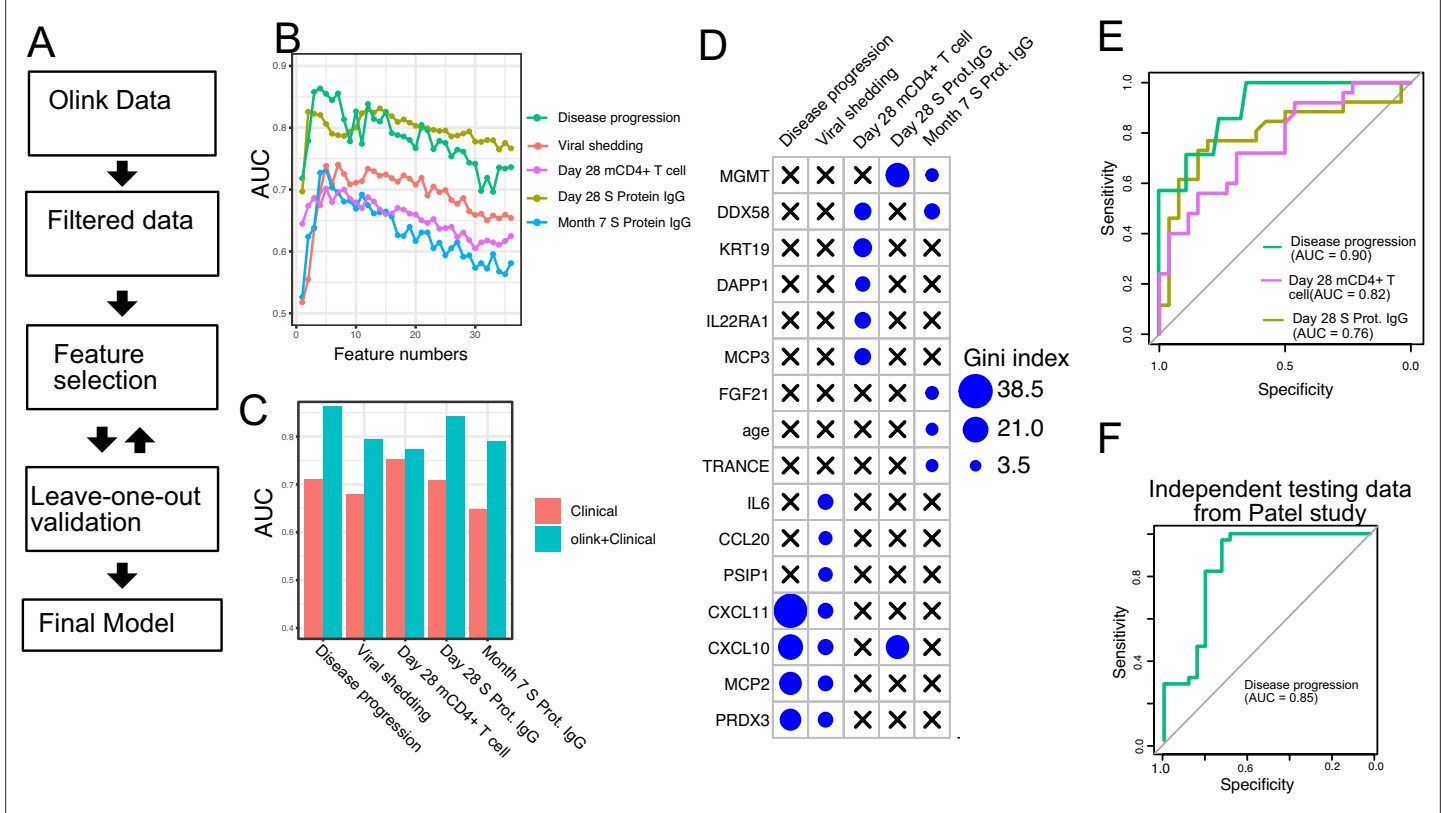

**Figure 7.** Plasma protein markers predict disease progression, T cell response, and spike (S) protein-binding IgG level in COVID-19 patients.
(**A**) Machine-learning procedure for predicting COVID-19 patient outcomes using Olink proteomics data collected at day 0. (**B**) Random forest models were built to predict symptom severity, S protein-binding IgG level at 28 days and 7 months after enrollment, and cytokine+ memory CD4+ T cells 28 days after enrollment. The plot shows the leave-one-out cross-validation performance (measured by AUC of the ROC) achieved by random forest models with different numbers of features. (**C**) The leave-one-out cross-validation performance of the best-performing models and the models using demographical data (age and sex) only. (**D**) Feature importance of the final random forest models for predicting symptom severity, S protein-binding IgG level at 28 days and 7 months after enrollment, and cytokine+ memory CD4+ T cells at 28 days after enrollment. (**E**) We generated an independent dataset using samples obtained from 64 COVID-19 patients enrolled in the placebo arm of a clinical trial of favipiravir. The performance of the machine-learning models was tested using the new dataset. (**F**) We used the final model to predict severe cases in an independent dataset and measured the performance of the model measured by the ROC curve.

The online version of this article includes the following figure supplement(s) for figure 7:

**Figure supplement 1.** We estimated the association between immune measures (blood transcription module [BTM] and Olink proteins measures) and COVID-19 outcomes using data from both lambda trial and an independent dataset from Favipiravir trial.

## Plasma proteins predict symptom severity, T cell response, and S protein-binding IgG levels in COVID-19 patients

Finally, we tested if plasma proteins measured at enrollment (day 0) can accurately predict disease progression, oropharyngeal viral load, and SARS-CoV-2-specific memory T cell and antibody responses manifested later in the study. We adopted a computational pipeline to select a small subset of predictive biomarkers from the 184 proteins measured by Olink assays. We used a leave-one-out cross-validation strategy to iteratively evaluate model performance and random forest for feature selection and for building the final model (*Figure 7A*). Based on results from cross-validation, we selected between 2 and 7 protein markers measured at early infection to predict each of the five outcomes. The final models achieved cross-validation AUC of 0.87, 0.78, 0.72, 0.81, and 0.77 for predicting disease progression, oropharyngeal viral load, day 28 SARS-CoV-2-specific CD4+ T cell responses, day 28 S protein-binding IgG levels and month 7 S protein-binding IgG levels, respectively (*Figure 7B*).

We compared the final models to baseline models that use only demographic (age and gender) and comorbidity (hypertension, diabetes, and obesity) data. The selected protein markers substantially improved the prediction of disease progression, S protein-binding IgG levels at day 28 and

month 7, and oropharyngeal viral load. On the other hand, protein markers did not improve the prediction for SARS-CoV-2-specific CD4+ T cell responses at day 28.

To validate the single-variable associations (*Figure 5*) and the multi-variable machine-learning models (*Figure 7A–D*), we generated an independent dataset using longitudinal, acute, and convalescent samples obtained from 54 COVID-19 participants enrolled in the placebo arm of an outpatient clinical trial of favipiravir (NCT04358549) (*Holubar et al., 2022*). Similar to participants in the Lambda trial, participants in the Favipiravir trial were recruited if they presented with initially mild to moderate COVID-19 at diagnosis, and the median duration of symptoms prior to randomization was 5 days, (*Supplementary file 5*). Among study participants, 7 of 54 (13%) later developed progressive and more severe symptoms and presented to the emergency department or were hospitalized (median 4 days to progression, range 2–10 days). We measured plasma proteomics by Olink at the time of enrollment (day 0) and neutralizing antibody levels and SARS-CoV-2-specific CD4+ T cell responses 28 days post-enrollment. Using this new dataset, we validated associations between early proteomic markers and longitudinal clinical and immunology outcomes (*Figure 7—figure supplement 1*). Importantly, we also demonstrate that machine-learning models using 2–7 plasma protein markers measured during acute infection and developed from the lambda dataset can accurately predict disease progression (AUC 0.90), SARS-CoV-2-specific CD4+ T cell responses (AUC 0.82), and the magnitude of S protein-binding IgG (AUC 0.76) at 28 days post-enrollment in the independent favipiravir dataset (*Figure 7E*).

We also tested if our model can accurately predict disease severity in a second independent dataset and identified a published dataset that characterized plasma proteins from 58 COVID-19 patients (26 moderate cases and 34 severe cases; *Patel et al., 2021*). Our model was able to accurately identify severe cases in the independent dataset, achieving an AUC of 0.85.

## Discussion

In this study, we longitudinally characterized the early immune response in patients who initially presented with mild to moderate COVID-19. With transcriptomic and proteomic profiling, we reveal a sequential activation of IFN signaling, T cells, and B cells within 2 weeks of symptom onset. We also identified associations between early immune profiles and later clinical, virological, and immunological outcomes. In particular, plasma RIG-I levels, early IFN signaling, and related cytokines (CXCL10, MCP1, MCP-2, and MCP-3) were associated with multiple patient outcomes, including disease progression, viral shedding, and the SARS-CoV-2-specific T cell and antibody response measured up to 7 months after enrollment. We observed that the immune response after the first dose of SARS-CoV-2 mRNA vaccination largely recapitulates the trajectory of immune response after SARS-CoV-2 infection, and associations between early proteomic signatures and adaptive immune responses were similar following natural infection and vaccination. Finally, we demonstrate that machine-learning models using 7–10 plasma protein markers are able to accurately predict disease progression, SARS-CoV-2-specific T cell magnitude, and the SARS-CoV-2 antibody response in independent datasets.

We found that variations in the early immune response following natural infection shape the long-term outcome of COVID-19 outpatients. In particular, early transcriptomic signatures of type I IFN and RIG-I signaling, as well as elevated levels of downstream, IFN-induced chemokines (CXCL10 and CXCL11), were associated with an increased risk of subsequent disease progression. These results are consistent with some previously reported studies (*Lucas et al., 2020*; *Yang et al., 2020*), but not others, which found that severe disease is associated with a defective IFN response (*Arunachalam et al., 2020*; *Zhang et al., 2020*). Some of these differences may be related to the timing of the assessment. Samples in our study were obtained in outpatients, prior to disease progression and hospitalization; in most other studies, patient samples were obtained at the time of hospitalization and/or when patients already had evidence of severe disease. This suggests a complex, non-monotonic relationship between the IFN response and disease severity.

Importantly, plasma levels of RIG-I, a protein not associated with time since symptom onset, were associated with all measured patient outcomes, including disease progression, oropharyngeal viral load, and SARS-CoV-2-specific T cell and antibody responses. RIG-I has been shown to be critically important in the response to several RNA viruses, including influenza virus, typically via interactions with the adapter protein mitochondrial antiviral-signaling protein (MAVS) and downstream type I and type III IFN upregulation. RIG-I was recently shown to play an important role in both sensing SARS-CoV-2 RNA and inhibiting SARS-CoV-2 replication in human lung cells but not via downstream MAVS

induction (*Yamada et al., 2021*). Rather, interactions between the RIG-I helicase domain and SARS-CoV-2 RNA induced an inhibitory effect on viral replication, independent of downstream IFN upregulation (*Yamada et al., 2021*). Our data showing an inverse association between plasma RIG-I levels and viral load are consistent with RIG-I having an important role in restricting early virus replication. As RIG-I is an intracellular RNA sensor, we sought to better understand why levels of RIG-I were elevated in the plasma by assessing associations with other transcriptomic modules and plasma proteins. Plasma RIG-I levels were modestly correlated with mRNA-level expression of RIG-I and RIG-I signaling modules and strongly correlated with plasma levels of several intracellular proteins, including DFFA, an intracellular protein known to be involved in cell death (*Zhang and Xu, 2000*), suggesting that plasma RIG-I levels may reflect both increased gene expression and increased cellular apoptosis. This hypothesis is consistent with a recent report which observed significant associations between gene expression signatures of apoptosis in plasmacytoid dendritic cells with increased disease severity (*Liu et al., 2021*).

We also observed that higher expression of three chemotactic receptor (CCR2) ligands MCP1, MCP2, and MPC3 were associated with disease progression. Higher plasma MCP3 levels have previously been shown to be elevated in SARS-CoV-2-infected patients with severe disease in comparison with those without (*Yang et al., 2020*), and transcriptome-wide association in lung tissue has found that higher expression of monocyte-macrophage CCR2, the receptor for MCP1 and MCP3, is associated with severe COVID-19 (*Pairo-Castineira et al., 2021*). However, we also observed that these CCR2 ligands are associated with positive virological and immunological outcomes, including reduced oropharyngeal viral load, and increased SARS-CoV-2-specific T cells and S protein-binding IgG levels. Consistent with a potential beneficial role, murine studies have found that CCR2 is essential for the survival of mice after pathogen challenge (*Pamer, 2009*; *Lim et al., 2011*; *Kurihara et al., 1997*). Taken together, these results demonstrated the complex role of CCR2 signaling in regulating immune responses. While essential for an effective immune response, overexpression may lead to severe symptoms and tissue damage. Therapeutic strategies to balance the positive and negative effects of CCR2 signaling may benefit the management of COVID-19 patients.

Our study design allowed us to compare immune responses following Peginterferon lambda treatment. Although there is literature that suggests IFN lambda-1a (IL-29) may impact immune function in vitro and in animal models (*Syedbasha and Egli, 2017*), the impact of therapeutic administration on immune function in humans remains unclear. In this study, we found that administration of Peginterferon lambda had no significant effect on the immunological profiles of participants. Specifically, we observed no significant impact of Peginterferon lambda on whole blood transcriptomic profiles 5 days post-administration or in SARS-CoV-2-specific T cell and antibody responses 28 days post-administration. Although we identified two plasma proteins elevated in Peginterferon lambda-treated individuals, these proteins were not associated with downstream patient outcomes. There are several possible reasons for these findings. First, the dosing used in this trial may not have achieved therapeutic levels in the upper respiratory epithelia, where its impact on immune cells might be expected to occur (*Jagannathan et al., 2021*). Second, the median symptom duration was 5 days at the time of randomization, and 40% of participants were already SARS-CoV-2 IgG positive at enrollment. It is possible that earlier administration, or prophylactic administration prior to established infection, may have had a different impact on immune outcomes. Arguing against this, in sensitivity analyses, we observed no difference in the treatment effect on immune profiles among individuals who were SARS-CoV-2 seronegative at enrollment.

Although naturally acquired SARS-CoV-2 infection results in protective antibody and T cell immune responses, reinfections can occur, and the precise determinants driving susceptibility to reinfection remain unclear (*To et al., 2021*). The BNT162b2 (Pfizer–BioNTech) vaccine has been shown to be highly effective in preventing SARS-CoV-2 infection (*Skowronski and De Serres, 2021*), although breakthrough cases have been increasingly reported since its approval (*CDC COVID-19 Vaccine Breakthrough Case Investigations Team, 2021*). Comparing the vaccine response with the immune response of natural infection may shed light on determinants of protective immunity to SARS-CoV-2 and potential ways to improve COVID-19 vaccines. Our analysis reveals that the proteomic response of the BNT162b2 vaccine mirrors in many ways the proteomic response after SARS-CoV-2 infection. Furthermore, several associations between early protein markers and protective immunity (T cell and antibody responses) were shared between COVID-19 patients and individuals who received the

BNT162b2 vaccine. These results suggest that plasma biomarkers might be useful to identify individuals at risk of SARS-CoV2 reinfection as well as breakthrough infection after vaccination.

Our study has some limitations. First, while we identified multiple associations between early immune measures and the outcome of COVID-19 patients, we did not establish causal relationships between them. Future studies are needed to perturb key immune modules in the early immune response and test their effect on the patient outcomes. Second, our study measured the immune response during the first 2 weeks after symptom onset in COVID-19 patients. Earlier immune responses between the initial infection and symptom onset have not been characterized. This is due to the difficulty to detect pre-symptomatic COVID-19 infection. Routine SARS-COV-2 monitoring in a select cohort will be required to acquire samples prior to and immediately after the infection in order to assess whether pre-infection signatures predict outcomes in COVID-19 patients. In addition, our trajectory analysis is based on the population-level data with only two time points sampled per patient. Further studies with more frequent sampling will be needed to confirm the immune trajectory at the individual level. Individuals in the lambda and favipiravir study were enrolled prior to the emergence and dominance of SARS-CoV-2 B.1.617 (Delta) and B.1.1.529 (Omicron) variants in the United States. Further validation studies are needed to test if the findings may be able to be generalized to infection with other variants. Finally, we used median values as cutoffs to identify individuals with high or low T cell and antibody responses when training the machine-learning model. However, the use of median cutoff is not ideal, as it does not reflect the minimal threshold of protective immunity. Further studies are needed to identify clinically relevant thresholds and to test the machine-learning models for predicting protection against reinfection.

Identification of patients at high risk of disease progression remains a critical need in management of patients with COVID-19. Several novel therapeutics have recently been issued emergency use authorization by the Federal Drug Administration, including monoclonal antibodies (e.g. casirivimab and imdevimab, sotrovimab), nucleoside inhibitors (molnupiravir), and protease inhibitors (nirmaltrevir/ritonavir). These therapeutics have been shown to be effective in reducing the risk of hospitalization or death among patients with high risk of disease progression in outpatient studies, but their supplies remain limited. Our data suggest that measurement of plasma proteins at the time of diagnosis could be a powerful adjunct to identify those patients who would most benefit from therapeutic interventions aimed at preventing progressive disease and hospitalization. Furthermore, our models can potentially be used to predict the degree of immune memory variation following natural infection and vaccination.

# Materials and methods

## Peginterferon lambda study design

Data and samples were obtained from a phase 2, single-blind, randomized placebo-controlled trial to evaluate the efficacy of lambda in reducing the duration of viral shedding in outpatients. The trial was conducted within the Stanford Health Care System, and participants were enrolled between April 25 and July 17, 2020. Adults aged 18–65 years with an U.S. Food and Drug Administration (FDA) emergency use authorized rRT-PCR positive for SARS-CoV-2 within 72 hr from swab to the time of enrollment were eligible for participation in this study. We included both symptomatic and asymptomatic patients based on the previous finding that the detected infectious virus were similar in samples from asymptomatic and symptomatic persons (*CDC COVID-19 Vaccine Breakthrough Case Investigations Team, 2021*). Symptomatic individuals were eligible given the presence of mild to moderate symptoms without signs of respiratory distress. Asymptomatic individuals were eligible if infection was the initial diagnosis of SARS-CoV-2 infection. Exclusion criteria included current or imminent hospitalization, respiratory rate >20 breaths per minute, room air oxygen saturation <94%, pregnancy or breastfeeding, history of decompensated liver disease, recent use of IFNs, antibiotics, anticoagulants, or other investigational and/or immunomodulatory agents for treatment of COVID-19, and prespecified lab abnormalities. Full eligibility criteria are provided in the study protocol. The protocol was amended on June 16, 2020 after 54 participants were enrolled but before results were available to include adults up to 75 years of age and eliminate exclusion criteria for low white blood cell and lymphocyte count. After confirming eligibility and providing informed consent in the patient's primary language, participants underwent a standardized history and physical exam and completed

bloodwork. If inclusion criteria were met, participants were enrolled. The trial was registered at ClinicalTrials.gov (NCT04331899) and approved by the Institutional Review Board of Stanford University, including approval for collection of biospecimens as per the study protocol (IRB 55619). Analysis of deidentified patient samples was approved by the Institutional Review Board of Stanford University (IRB 57230) and University of California San Francisco (UCSF) (Stanford IRB reliance).

The patients were randomly assigned to lambda or placebo in a 1:1 ratio to the treatment and control arms using a computer-generated randomization scheme developed by the study data management provider. The study data management provider completed a password-protected electronic spreadsheet containing the randomization allocation, along with the code used to generate the allocation and the seed used in the random number generation. These are stored on secure servers at Stanford. No randomization was applied during the bioinformatics analysis of the data.

### Participant follow-up and sample collection

Participants completed a daily symptom questionnaire using REDCap Cloud version 1.5. In-person follow-up visits were conducted at days 1, 3, 5, 7, 10, 14, 21, and 28, with assessment of symptoms and vitals, and collection of oropharyngeal swabs (FLOQ Swabs; Copan Diagnostics). Peripheral blood was collected at enrollment, day 5, day 28, and month 7 post-randomization. Whole blood was collected in Paxgene Tubes, and remaining blood was processed for plasma and peripheral blood mononuclear cells.

### Clinical laboratory procedures

Laboratory measurements were performed by trained study personnel using point-of-care Clinical Laboratory Improvement Amendments (CLIA)-waived devices or in the Stanford Health Care Clinical Laboratory. Oropharyngeal swabs were tested for SARS-CoV-2 in the Stanford Clinical Virology Laboratory using an emergency use authorized, laboratory-developed RT-PCR. Centers for Disease Control and Prevention guidelines identify oropharyngeal swabs as acceptable upper respiratory specimens to test for the presence of SARS-CoV-2 RNA, and detection of SARS-CoV-2 RNA swabs using oropharyngeal swabs was analytically validated in the Stanford Virology Laboratory.

### Clinical and virological outcome definitions

Clinical and virological outcomes assessed in this study included those reported in the main clinical trial (statistical analysis plan at https://exhibits.stanford.edu/data/catalog/hc972ys6733). Disease progression was defined by incident emergency department and/or hospitalizations within 28 days of enrollment in the study and was a prespecified secondary outcome of the clinical trial. Viral shedding was assessed as SARS-CoV-2 oropharyngeal viral RNA AUC. To calculate viral AUC, we identified the cycle threshold (Ct) value using the fluorescence vs. cycle data reported from RT-PCR scanner. Ct values were subtracted from the detect limit (Ct = 41) to quantify the viral shedding in each oropharyngeal (OP) swab. We plotted the viral shedding in each visit vs. time and calculated the AUC using numerical integration based on the trapezoid rule.

### Whole blood transcriptomics

Whole blood transcriptomics were performed at Novogene Corporation, Inc Briefly, whole blood samples collected in Paxgene Tubes were first treated with proteinase K and then RNA extraction performed using Quick-RNA MagBead Kit (R2132) on KingFisher followed by sample quality control checks using a Qubit and Bioanalyzer 2100. Libraries were prepared using ZymoSeq RiboFree Total RNA Library Kit (R3000). Sequencing took place on a Nova6000 on an S4 lane, 30 M paired reads, PE 150.

### Whole blood transcriptomic data analysis

The transcript-level count data and transcript per million data were calculated using Kallisto (*Bray et al., 2016*) (v0.46.2) and human cDNA index produced using Kallisto on Ensembl v96 transcriptomes. For each RNA-seq sample, we calculated the single-sample enrichment score of the BTM using the fgsea R package (*Korotkevich et al., 2021*). The enrichment scores of the BTMs were normally distributed across samples and are treated as variables, similar to individual protein markers, in the downstream analysis.

## Plasma protein profiling using Olink panels

We measured proteins in plasma using Olink multiplex proximity extension assay (PEA) inflammation panel and immune response panel (Olink proteomics, https://www.olink.com/) according to the manufacturer's instructions. The PEA is a dual-recognition immunoassay, where two-matched antibodies labeled with unique DNA oligonucleotides simultaneously bind to a target protein in solution. This brings the two antibodies into proximity, allowing their DNA oligonucleotides to hybridize, serving as a template for a DNA polymerase-dependent extension step. This creates a dsDNA 'barcode' unique for the specific antigen and quantitatively proportional to the initial concentration of target protein. The hybridization and extension are immediately followed by PCR amplification, and the amplicon is then finally quantified by microfluidic qPCR.

## T cell assays

SARS-CoV-2-specific T cell peptide pools were purchased from Miltenyi Biotec (PepTivator SARS-CoV-2 Prot_S, Prot_S1, Prot N, and Prot M) and resuspended in dimethyl sulfoxide (DMSO). These PepTivator reagents are pools of lyophilized peptides of 15 amino acid length with 11 amino acid overlap, covering immunodominant sequence domains of the S (S and S1) (aa sequence 1–1273), nucleocapsid (N), or membrane (M) proteins of SARS-CoV-2.

Antigen-specific T cell responses were measured using an intracellular cytokine staining assay. Briefly, cryopreserved PBMCs were thawed, counted, and resuspended in complete RPMI (RPMI [Corning] supplemented with 10% Fetal bovine serum (FBS) (Gibco), 100 IU penicillin [Corning], 100 μg/ml streptomycin [Corning], 1 mM Hepes [Corning], and 2 mM L-glutamine [Corning]). The cells were plated in 96-well U bottom plates at 1x10e6 PBMCs per well and then rested overnight at 37°C in a CO2 incubator. The following morning, cells were cultured in presence of either SARS-CoV-2 peptides (1 μg/ml), Phorbol myristate acetate (PMA) (300 ng/ml), and Ionomycin (1.5 μg/ml) as positive control or media as a negative control for 6 hr at 37°C. All conditions were in the presence of brefeldin A (BD Pharmingen), monensin (BD Pharmingen), and CD107a. After a 6 hr incubation, cells were washed and surface stained for CCR7 for 15 min at 37°C, followed by the remaining surface stain for 30 min at room temperature (RT) in the dark. Thereafter, cells were washed twice with PBS containing 0.5% BSA and 2 mM EDTA, then fixed/permeabilized (FIX & PERM Cell Permeabilization Kit, Invitrogen) and stained with intracellular antibodies for 20 min at RT in the dark. A complete list of antibodies is listed in supplementary methods. All samples were analyzed on an Attune NXT flow cytometer and analyzed with FlowJo X (Tree Star) software.

## Antibody assays

IgG antibody titers against the SARS-CoV-2 full-length S protein were assessed by enzyme-linked immunosorbent assay (ELISA) (*Chakraborty et al., 2022*). Briefly, 96-well half-area microplates (Corning [Millipore Sigma]) plates were coated with antigens at 2 μg/ml in PBS for 1 hr at RT. Next, the plates were blocked for an hour with 3% non-fat milk in PBS with 0.1% Tween 20 (PBST). Plasma was diluted fivefold starting at 1:50 in 1% non-fat milk in PBST. 25 μl of the diluted plasma was added to each well and incubated for 2 hr at RT. Following primary incubation, 25 μl of 1:5000 diluted horse radish peroxidase conjugated anti-human IgG secondary antibodies (Southern Biotech) were added and incubated for 1 hr at RT. The plates were developed by adding 25 μl/well of the chromogenic substrate 3,3',5,5'-tetramethylbenzidine solution (Millipore Sigma). The reaction was stopped with 0.2 N sulphuric acid (Sigma), and absorbance was measured at 450 nm (iD5 SPECTRAmax, Molecular Devices). The plates were washed five times with PBST between each step, and an additional wash with PBS was done before developing the plates. All data were normalized between the same positive and negative controls, and the binding AUC were calculated using GraphPad PRISM (Version 9).

## Analysis of Olink data from the vaccine study

The Olink data from the mRNA vaccine study was previously obtained and published (*Goyal et al., 2020*). We tested if the level of the proteins are significantly altered after vaccination using ANOVA (expression ~time), where time is treated as a categorical variable to account for non-linear behavior of the proteins. p Values from the ANOVA models are adjusted using the FDR method (*Benjamini and Hochberg, 1995*). To visualize the trajectory of the proteins, we imputed the protein level in each day using linear interpolation with the 'approx' function in R.

## Favipiravir validation study

As a validation study, we utilized samples collected from participants enrolled in a similarly designed, phase 2, double-blind randomized placebo-controlled trial of favipiravir in mildly symptomatic or asymptomatic adults with a positive SARS-CoV-2 RT-PCR assays within 72 hr of enrollment. The trial was conducted within the Stanford Health Care System, and participants were enrolled between July 8, 2020 and March 23, 2021. Informed consent was obtained from all participants prior to enrollment in the study per IRB guidelines. Clinical outcomes assessed include incident emergency room (ER) visits and hospitalizations (prespecific secondary outcomes in primary study). Only samples from placebo participants were included in this analysis. Peripheral blood was collected at enrollment and day 28 post-randomization and processed for plasma and peripheral blood mononuclear cells. Day 0 plasma proteomics by Olink and day 28 T cell assays were performed as described above. Anti-SARS-CoV-2 serology was performed using a virus plaque reduction neutralization assay (Viroclinics Biosciences, Rotterdam, The Netherlands [*Holubar et al., 2022*]). The study was performed as an investigator-initiated clinical trial (NCT 04346628) and approved by the Institutional Review Board of Stanford University including approval for collection of biospecimens as per the study protocol (IRB 56032). Analysis of deidentified patient samples was approved by the Institutional Review Board of Stanford University (IRB 57230) and UCSF (Stanford IRB reliance).

## Statistical analysis

In the parent clinical trial from which samples were obtained, sample size was estimated based on the primary outcome measured in that study – oropharyngeal viral load shedding cessation. Assuming 1:1 randomization and the use of a two-sided log rank test at the alpha = 0.04999 level of significance for the final analysis, we anticipated the occurrence of 79 shedding cessation events, which provided 80% power to detect a hazard ratio of 2.03. We additionally assumed median time to shedding cessation of 14 days in the control arm and 7 days in the treatment arm, a 2-month accrual period, a 2-week follow-up period after randomization of the last patient, and 10% drop out in the control arm. This enabled an interim analysis conducted at alpha = 0·00001 to assess overwhelming efficacy after 50% of participants completed 24 hr of follow-up. We estimated that the total sample size required to achieve 79 events was 120 (60 participants per arm). For this secondary analysis of transcriptomic and proteomic data, all subjects with available data for analysis were included.

PCA was conducted by applying the prcomp function in base R to the whole Olink dataset or the top 500 genes with the highest variance. To access the association between the PCs and clinical data, we fitted regression models (PC ~clinical variable). The percent of variances explained by the clinical variable is used to measure the association.

We accessed the association between the expression of immune modules or protein markers with time using the regression model (expression ~treatment + time + $time^2$). Because the treatment only affects samples collected at day 5 but not day 0, we assigned the treatment variable as 1 to day 5 sample from the treatment arm and as 0 to other samples in the regression analysis. Goodness of fit analysis using Bayesian information criterion (BIC) shows that polynomial terms with orders higher than two do not improve the model. It should be noted that our study contains repeated measures of the same individuals in two time points (0 and 5 days after enrollment). While including subjects as random effects in the regression model allows the model to adjust for individual differences, it resulted in near-singular fits of the data for many of the immune measurements. To avoid model over-fitting, we decided to only include the fixed effects (time) in our model. To find significant associations, we compared the model with the base model (expression ~1) and used the F test to calculate the p value. We adjusted the p value using the FDR method. We performed a parallel analysis using mixed-effect models (expression ~treatment time + $time^2$+subject ID [random effect]) to fit the data and found that all significant (FDR <0.05) variables identified using the fixed effect model were also significant in the mixed-effect model (*Supplementary file 2*).

We estimated the enrichment score of the major immune cell types using the xCell package (*Aran et al., 2017*). The association between the xCell scores and time was tested using the same regression method described above.

We tested the association between immune measurements and disease progression using regression models (measurements ~symptom severity +treatment + time + $time^2$) and the lm function in R. Because the treatment only affects samples collected at day 5 but not day 0, we assigned the treatment

variable as '1' in day 5 samples from the treatment arm and as '0' in other samples in the regression analysis. The p value of the symptom severity term is adjusted using the FDR method. Similar regression models were used to test the association between immune measurements and other outcomes, including the oropharyngeal viral load, the memory CD4+ T cell activity and S protein-binding IgG levels. To test between immune measurements and symptom severity without adjusting the time to symptom onset, we performed a one-way anova analysis using the lm function in R (measurements ~symptom severity).

## Predictive modeling

We used the protein measurements (measured by Olink assays) to predict the clinical, virological, and memory T cell activity and IgG antibodies. Since the outcomes are a mixture of categorical (symptom severity) and continuous (viral load, T cell, and antibody responses) variables, we framed all prediction tasks as classification problems by dichotomizing the continuous variables using median as cutoffs. To ensure Olink data were measured before clinical progression, we excluded Olink data that were collected at day 5. To prevent overfitting, we selected 30 proteins with the highest variance as input data, as the highly variable proteins best capture the inter-subject difference across the COVID-19 patients. We further select features using random forest and leave-one-out validation. In step 1, we train a random forest model using data from all samples but one left out sample. In step 2, we rank the feature importance of the 30 protein markers based on the gini index reported by the random forest model. In step 3, we train reduced random forest models with 1–29 most important proteins. In step 4, we predict the outcomes using the data from the left-out sample. We repeat steps 1–4 until we predict the outcome of all samples. We calculate the model performance using the AUROC curve (AUC). The variable combinations that give rise to the highest AUC are selected as the optimal model. The optimal model for predicting symptom severity was tested using Olink data from two independent studies.

## Data availability

The RNA-sequencing data is available in GEO under the accession number GSE178967. The Olink, clinical, virological, and immunological, as well as the machine-learning models and the source codes, are available in the github repository: (https://github.com/hzc363/COVID19_system_immunology, copy archived at swh:1:rev:3a58314134c2c117d1e4989d9cecec243f134dbd; *Hu, 2022*).

## Acknowledgements

Support for the study was provided from NIH/NIAID (U01 AI150741-01S1 to ZH, ST, IRB, BG, TW, and PJ), the Stanford's Innovative Medicines Accelerator, and NIH/NIDA DP1DA046089. The Lambda clinical trial was funded by anonymous donors to Stanford University, and Peginterferon Lambda provided by Eiger BioPharmaceuticals. The funders had no role in data collection and analysis or the decision to publish.

We thank all study participants who participated in this study, the study team for their tireless work, and Thanmayi Ranganath, Nancy Q Zhao, Aaron J Wilk, Rosemary Vergara, Julia L McKechnie, Giovanny J Martínez-Colón, Arjun Rustagi, Geoff Ivison, Ruoxi Pi, Madeline J Lee, Taylor Hollis, Georgie Nahass, Kazim Haider, and Laura Simpson for assistance with processing samples. We also thank our colleagues at Stanford University Occupational Health and at San Mateo Medical Center for participant referrals. The Stanford REDCap platform (http://redcap.stanford.edu) is developed and operated by Stanford Medicine Research IT team. The REDCap platform services at Stanford are subsidized by (a) Stanford School of Medicine Research Office, and (b) the National Center for Research Resources and the National Center for Advancing Translational Sciences, National Institutes of Health, through grant UL1 TR001085.

## Additional information

### Competing interests

Isabel Rodriguez-Barraquer: Reviewing editor, *eLife*. The other authors declare that no competing interests exist.

## Funding

| Funder | Grant reference number | Author |
|---|---|---|
| National Institute of Allergy and Infectious Diseases | U01 AI150741-01S1 | Zicheng Hu<br>Kattria van der Ploeg<br>Saki Takahashi<br>Isabel Rodriguez-Barraquer<br>Bryan Greenhouse<br>Atul J Butte<br>Taia T Wang<br>Prasanna Jagannathan |
| Stanford University | Stanford's Innovative Medicines Accelerator | Haley Hedlin<br>Bali Pulendran<br>Prasanna Jagannathan |
| National Institute on Drug Abuse | DP1DA046089 | Catherine Blish |
| National Institute of Allergy and Infectious Diseases | T32-AI052073 | Karen B Jacobson |

The funders had no role in study design, data collection and interpretation, or the decision to submit the work for publication.

## Author contributions

Zicheng Hu, Conceptualization, Software, Formal analysis, Validation, Investigation, Visualization, Methodology, Writing – original draft, Writing – review and editing; Kattria van der Ploeg, Conceptualization, Data curation, Validation, Investigation, Methodology; Saborni Chakraborty, Conceptualization, Data curation, Writing – original draft, Writing – review and editing; Prabhu S Arunachalam, Marisa Holubar, Kathleen Press, Resources, Data curation; Diego AM Mori, Hector Bonilla, Julie Parsonnet, Aruna Subramanian, Chaitan Khosla, Yvonne Maldonado, Lauren de la Parte, Maureen Ty, Data curation; Karen B Jacobson, Data curation, Funding acquisition; Jason R Andrews, Upinder Singh, Conceptualization, Funding acquisition; Haley Hedlin, Formal analysis, Methodology; Gene S Tan, Saki Takahashi, Isabel Rodriguez-Barraquer, Methodology; Catherine Blish, Funding acquisition, Methodology; Bryan Greenhouse, Supervision, Funding acquisition, Methodology; Atul J Butte, Supervision, Funding acquisition; Bali Pulendran, Resources; Taia T Wang, Conceptualization, Funding acquisition, Methodology; Prasanna Jagannathan, Conceptualization, Formal analysis, Supervision, Funding acquisition, Writing – original draft, Project administration, Writing – review and editing

## Author ORCIDs

Zicheng Hu http://orcid.org/0000-0002-4168-1725
Kattria van der Ploeg http://orcid.org/0000-0002-3183-3190
Karen B Jacobson http://orcid.org/0000-0003-3050-4595
Julie Parsonnet http://orcid.org/0000-0001-7342-5366
Jason R Andrews http://orcid.org/0000-0002-5967-251X
Haley Hedlin http://orcid.org/0000-0002-0757-7959
Isabel Rodriguez-Barraquer http://orcid.org/0000-0001-6784-1021
Bryan Greenhouse http://orcid.org/0000-0003-0287-9111
Atul J Butte http://orcid.org/0000-0002-7433-2740
Upinder Singh http://orcid.org/0000-0003-0630-0306
Bali Pulendran http://orcid.org/0000-0001-6517-4333
Prasanna Jagannathan http://orcid.org/0000-0001-6305-758X

## Ethics

Clinical trial registration NCT04331899.

Human subjects: The Peginterferon Lambda study was performed as an investigator-initiated clinical trial (NCT 04331899), and approved by the Institutional Review Board of Stanford University, including approval for collection of biospecimens as per the study protocol (IRB 55619). The Favipiravir study was performed as an investigator-initiated clinical trial (NCT 04346628), and approved

by the Institutional Review Board of Stanford University including approval for collection of biospecimens as per the study protocol (IRB 56032). In both studies. informed consent was obtained from all participants prior to enrollment in the per IRB guidelines. Analysis of deidentified patient samples collected from the Peginterferon Lambda and Favipiravir studies was approved by the Institutional Review Boards of Stanford University (IRB 57230) and UCSF (Stanford IRB reliance).

## Decision letter and Author response

Decision letter https://doi.org/10.7554/eLife.77943.sa1
Author response https://doi.org/10.7554/eLife.77943.sa2

## Additional files

### Supplementary files
- Supplementary file 1. Characteristics of study participants (lambda study).
- Supplementary file 2. Statistical analysis of the trajectory analysis of early stage immune measures in COVID-19 patients.
- Supplementary file 3. Association between early stage immune measures in COVID-19 patients and their clinical outcomes.
- Supplementary file 4. Intracellular cytokine staining (ICS) antibody panel.
- Supplementary file 5. Characteristics of validation study participants (placebo arm and Favipiravir trial).
- MDAR checklist

### Data availability
The RNA-sequencing data is available in GEO under the accession number GSE178967. Other data and computational models are deposited at GitHub (https://github.com/hzc363/COVID19_system_immunology, copy archived at swh:1:rev:3a58314134c2c117d1e4989d9cecec243f134dbd).

The following datasets were generated:

| Author(s) | Year | Dataset title | Dataset URL | Database and Identifier |
|---|---|---|---|---|
| Hu Z, van der Ploeg K, Chakraborty S, Arunachalam P, Mori D, Jacobson K, Jagannathan P | 2022 | RNA-seq data of samples from COVID-19 patients | https://www.ncbi.nlm.nih.gov/geo/query/acc.cgi?acc=GSE178967 | NCBI Gene Expression Omnibus, GSE178967 |
| Hu Z, van der Ploeg K, Chakraborty S, Arunachalam P, Mori D, Jacobson K, Jagannathan P | 2022 | Olink and clinical data of samples from COVID-19 patients | https://github.com/hzc363/COVID19_system_immunology | GitHub, hzc363/COVID19_system_immunology |

The following previously published dataset was used:

| Author(s) | Year | Dataset title | Dataset URL | Database and Identifier |
|---|---|---|---|---|
| Patel H, Ashton N J, Dobson R J, Andersson L M, Yilmaz A, Blennow K, Zetterberg H | 2021 | Proteomic blood profiling in mild, severe and critical COVID-19 patients | https://doi.org/10.5281/zenodo.3895886 | Zenodo, 10.5281/zenodo.3895886 |

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
