## [Editor Report]

This manuscript uses a multi-omics approach to investigate how early immune markers in blood predict subsequent clinical outcome and immune responses. The study uses samples from a previous trial and identifies several immune markers associated with later clinical and immunological outcomes in this cohort. An important next step will be to validate this in other cohorts and test the utility of this in a clinical setting.

---

## [Decision Letter]

**Decision letter after peer review:**

Thank you for submitting your article "Early immune responses have long-term associations with clinical, virologic, and immunologic outcomes in patients with COVID-19" for consideration by *eLife*. Your article has been reviewed by 2 peer reviewers, and the evaluation has been overseen by a Reviewing Editor and Miles Davenport as the Senior Editor. The reviewers have opted to remain anonymous.

Essential revisions:

One concern is that the two groups in a clinical trial with an immunomodulatory drug were combined in this analysis. The authors take for granted that the administration of Peg-IFN λ does not modify the course of the disease and therefore that treated and untreated individuals can be analyzed together. This is at odds with other randomized studies, that have shown antiviral and clinical effect of IFN-based therapy. In particular, Peg-λ accelerated viral decline in outpatients and prevented clinical deterioration in a study performed in a similar setting using the same dose than here (Feld et al., Lancet Resp Med 2021). Other positive results in early patients were found with IFN-β (Monk et al., Lancet Resp Med 2020). Even if the administration of IFN in this study had no clinical or virological benefits, it could nonetheless alter the kinetics of ISG. The authors claim it is not the case, but it is difficult to assess it based on the figures shown, using PCA.

Although PegIFN was not associated with significant changes in clinical outcomes, the lack of overall clinical responses was likely due to a mix of responders and non-responders. Were there variables at baseline identified by computational analysis which could predict responses to IFN? Given these difficulties, we would encourage the authors to do a separate analysis, which is a revision.

If the authors decide to continue with this revision we would also encourage them to address the following comments:

1. Add essential information on the clinical trial should be included in the manuscript to suggest whether research findings mostly apply to the β, δ or omicron era. All in all, since this study focuses on the host, findings should be generalizable irrespective of the pathogen.

2. It is questionable whether a strong claim can be made on disease progression, since only 8 patients were hospitalized in this study. In addition, it should be clarified when these patients progressed. Page 10, it is said that the median time to progression is 2 days, so in fact, the data collected at day 0 and 5 are very close, or even perhaps posterior to hospitalization in some cases, making it difficult to claim that it can be used for prediction. More generally data used are up to 14 days post symptom onset, while the median time to hospitalization in these populations is roughly ~8 days. This makes it here as well difficult to really argue that the model has a "predictive" value to anticipate disease progression.

3. If data used in the study are close to hospitalization, then this really diminishes the novelty of the findings, as many studies have already reported an association between these markers and disease severity (see also Young et al., Viral Dynamics and Immune Correlates of Coronavirus Disease 2019 (COVID-19) Severity, CID 2021).

4. The definition of disease progression seems to differ from the original study "Overall, 17 participants had evidence of disease progression, defined as hospitalization, presentation to the emergency department, or worsening cough or shortness of breath defined as an increase in severity of two points or more on a five-point scale"? Please clarify what is your endpoint and why, if relevant, it differs from the original study.

5. Can you clarify how viral shedding was analyzed? The fact that viral load is analyzed with a different metrics than other proteins when looking at predictors of disease progression is puzzling. Figure S5 does not seem to be convincing, which relies on AUC of viral load calculated in patients with high heterogeneity in their symptom onset. Please use the same approach for viral load than what was used for IP-10 in order to demonstrate that IP-10 is a better predictor of disease progression than viral load.

6. Regarding prediction, it is really unclear how the model using demographics was built. It is obvious than many other factors than age and sex are highly predictive of disease progression.

7. If you want these results to be useful for the clinical community then the model used in figure 7 should be explicitly given so that anyone can use these results to build score on its own population.

8. Improve discussion on IFN-λ and propose better graphs to justify the absence of effect of treatment. It would be more helpful to provide simple graphs, such as boxplots of the changes in the 7-10 relevant markers between day 0 and day 5 in treated and untreated individuals (along with p-values), so that the lack of difference can be easily visualized.

9. To build a better model of demographics, consider exploring more covariates. For instance, evaluate models including covariates that are significant in univariate analysis and that have a >10% prevalence (hypertension, diabetes, age>55…).

---

## [Author Response]

Essential revisions:One concern is that the two groups in a clinical trial with an immunomodulatory drug were combined in this analysis. The authors take for granted that the administration of Peg-IFN λ does not modify the course of the disease and therefore that treated and untreated individuals can be analyzed together. This is at odds with other randomized studies, that have shown antiviral and clinical effect of IFN-based therapy. In particular, Peg-λ accelerated viral decline in outpatients and prevented clinical deterioration in a study performed in a similar setting using the same dose than here (Feld et al., Lancet Resp Med 2021). Other positive results in early patients were found with IFN-β (Monk et al., Lancet Resp Med 2020). Even if the administration of IFN in this study had no clinical or virological benefits, it could nonetheless alter the kinetics of ISG. The authors claim it is not the case, but it is difficult to assess it based on the figures shown, using PCA.Although PegIFN was not associated with significant changes in clinical outcomes, the lack of overall clinical responses was likely due to a mix of responders and non-responders. Were there variables at baseline identified by computational analysis which could predict responses to IFN? Given these difficulties, we would encourage the authors to do a separate analysis, which is a revision.

We thank the reviewers for raising the concern regarding the combined analysis of the placebo control and Peginterferon Λ arms. We have addressed this concern in several additional ways in the revised manuscript.

First, in our revised manuscript, we fitted the trajectory of the immune measures using data from each individual arm and from both arms together (Figure 3A). We found that immune trajectories were similar between the two individual arms, suggesting that Peginterferon Λ treatment did not significantly affect early immune trajectories after COVID-19 infection.

Second, we reorganized out manuscript to highlight detailed analyses that we previously performed to compare responses between groups. We tested the effect of Peginterferon Λ on individual immune measures measured at day 5 post-treatment. Blood transcriptomic profiles were similar between the two groups, and only two proteins were significantly different between the Peginterferon Λ and placebo treatment arm (LAMP3 and HSD11B1, previously in supplemental, now moved to Figure 2I). However, neither LAMP3 nor HSD11B1 were associated with patient outcomes (hospitalization, viral load, T cell response, and antibody levels) in our later analysis using combined data or control-only data (Supplementary File 3).

Third, we have now included ‘treatment’ as a variable in our multi-variable regression analysis to reflect the two-arm design of the study and to control for potential differences between the two arms. See the method section for more details. Adding the treatment variable resulted in small changes to our results, but the most significant proteins associated with clinical outcomes remained the same. We have updated the results in the manuscript (Figure 3, Figure 4, Figure 5, Supplementary file 2 and Supplementary file 3).

Finally, we have included a new supplemental figure (Figure 5 —figure supplement 4) to compare the change of 20 most relevant (the same set of proteins as shown in Figure 5B) markers between day 0 and day 5 in placebo and Peginterferon Λ-treated individuals. This analysis revealed no significant change in these 20 plasma markers between day 0 and day 5 after multiple comparison correction.

Importantly, we previously characterized the association between immune measures and clinical outcomes (disease progression, viral load, T cell response, and antibody levels) using data from participants in the placebo arm only. We found that the results were consistent when compared with the analysis that included all participants (Figure 5 —figure supplement 3). We also validated associations and predictive models between early plasma proteins and patient outcomes using samples collected from placebo arm participants enrolled in a separate clinical trial of Favipiravir. We show that results from the Peginterferon Λ trial were consistent with the result from the Favipiravir trial (Figure 7 —figure supplement 1) and that early plasma proteins were highly predictive in this second dataset, which only includes placebo-treated individuals.

If the authors decide to continue with this revision we would also encourage them to address the following comments:1. Add essential information on the clinical trial should be included in the manuscript to suggest whether research findings mostly apply to the β, δ or omicron era. All in all, since this study focuses on the host, findings should be generalizable irrespective of the pathogen.

All three cohorts were enrolled prior to the emergence and dominance of SARS-CoV-2 B.1.617 (Δ) and B.1.1.529 (Omicron) variants in the United States. This has been added in the limitations section of the discussion.

2. It is questionable whether a strong claim can be made on disease progression, since only 8 patients were hospitalized in this study. In addition, it should be clarified when these patients progressed. Page 10, it is said that the median time to progression is 2 days, so in fact, the data collected at day 0 and 5 are very close, or even perhaps posterior to hospitalization in some cases, making it difficult to claim that it can be used for prediction. More generally data used are up to 14 days post symptom onset, while the median time to hospitalization in these populations is roughly ~8 days. This makes it here as well difficult to really argue that the model has a "predictive" value to anticipate disease progression.3. If data used in the study are close to hospitalization, then this really diminishes the novelty of the findings, as many studies have already reported an association between these markers and disease severity (see also Young et al., Viral Dynamics and Immune Correlates of Coronavirus Disease 2019 (COVID-19) Severity, CID 2021).

Response to point 2 and 3: we thank the reviewer for bring up this issue. Indeed, it is critical to examine if our model can predict disease progression before it occurs. In the revised manuscript, we updated our machine learning analysis to only use data from day 0 of the Peginterferon Λ trial. At day 0, all patients recruited had mild to moderate symptoms, and none of the patients displayed severe symptoms. We show that our machine learning model was able to accurately predict disease progression before it occurs (Figure 7).

The model was also able to predict disease progression of COVID-19 patients in a validation dataset (Favipiravir trial), where samples were similarly collected at Day 0. The median time to progression is 4 days (with a range of 2 to 10 days) in the Favipiravir trial, demonstrating that the model is able to predict disease progression beyond 2 days. We have included the time to progression information of the Favipiravir trial in our revised manuscript.

4. The definition of disease progression seems to differ from the original study "Overall, 17 participants had evidence of disease progression, defined as hospitalization, presentation to the emergency department, or worsening cough or shortness of breath defined as an increase in severity of two points or more on a five-point scale"? Please clarify what is your endpoint and why, if relevant, it differs from the original study.

In the original study, the incidence of hospitalizations and ER visits was a prespecified secondary analysis (see Statistical analysis plan published at: https://exhibits.stanford.edu/data/catalog/hc972ys6733). The outcome of disease progression noted by the reviewer that included worsening cough or shortness of breath was an exploratory analysis (also pre-specified in the statistical analysis plan). We chose the more standard outcome definition of disease progression to include incident hospitalization and ER visits in this analysis, and, importantly, this was also the same secondary clinical outcome as used in the Favipiravir study (the alternative outcome was not used in the Favipiravir study). This has been clarified in the methods, and additional details regarding the Favipiravir validation study have now been provided in the methods.

5. Can you clarify how viral shedding was analyzed? The fact that viral load is analyzed with a different metrics than other proteins when looking at predictors of disease progression is puzzling. Figure S5 does not seem to be convincing, which relies on AUC of viral load calculated in patients with high heterogeneity in their symptom onset. Please use the same approach for viral load than what was used for IP-10 in order to demonstrate that IP-10 is a better predictor of disease progression than viral load.

The viral shedding data were measured and analyzed as described in a previous publication (PMID: 33785743). We identified the cycle threshold (Ct) value using the fluorescence vs cycle data reported from RT-PCR scanner. We subtract the Ct value from the detect limit (Ct=41) to quantify the viral shedding in each OP swab. We plotted the viral shedding in each visit versus time and calculated the area under the curve using numerical integration based on the trapezoid rule.

To illustrate the necessity of using AUC rather than individual readouts of the Ct value for quantifying viral shedding, we have plotted the Ct trajectories in Author response image 1. Because of the sparse nature of the measurement, the Ct value from any single sample are not reliable to quantify the viral shedding of the patients. We decided to use the area under the curve because it captures both the intensity and the length of the viral shedding, both of which contribute to the spread of the COVID-19. In addition, we have included the symptom onset time as a variable in our regression analysis, which allows us to control for the heterogeneity in symptom onset of patients.

**Author response image 1. sa2fig1:** 

6. Regarding prediction, it is really unclear how the model using demographics was built. It is obvious than many other factors than age and sex are highly predictive of disease progression.

We updated our machine learning analysis and included comorbidities (diabetes, hypertension and obesity) in the dataset. We found that the “olink + demographic + comorbidity” model outperforms the baseline “demographic + comorbidity” model (Figure 7C). We have included the new result in the revised manuscript.

7. If you want these results to be useful for the clinical community then the model used in figure 7 should be explicitly given so that anyone can use these results to build score on its own population.

We have shared our random forest model in GitHub (https://github.com/hzc363/COVID19_system_immunology). In the repository, we have now included a sample dataset and a tutorial to demonstrate how the random forest models can be used for prediction.

8. Improve discussion on IFN-λ and propose better graphs to justify the absence of effect of treatment. It would be more helpful to provide simple graphs, such as boxplots of the changes in the 7-10 relevant markers between day 0 and day 5 in treated and untreated individuals (along with p-values), so that the lack of difference can be easily visualized.

We have included a new supplemental figure (Figure 5 —figure supplement 4) to compare the change of 20 most relevant (the same set of proteins as shown in Figure 5B) markers between day 0 and day 5 in placebo and Peginterferon Λ-treated individuals along with p-values.

9. To build a better model of demographics, consider exploring more covariates. For instance, evaluate models including covariates that are significant in univariate analysis and that have a >10% prevalence (hypertension, diabetes, age>55…).

As mentioned above, we tested the model with comorbidities (diabetes, hypertension and obesity) along with age and gender. We found that the “olink + demographic + comorbidity” model outperforms the baseline “demographic + comorbidity” model (Figure 7C). We have included the new result in the revised manuscript.